# Abstract Reward Processes: Leveraging State Abstraction for Consistent Off-Policy Evaluation

**Shreyas Chaudhari**
University of Massachusetts
schaudhari@cs.umass.edu

**Ameet Deshpande**
Princeton University
asd@cs.princeton.edu

**Bruno Castro da Silva**
University of Massachusetts
bsilva@cs.umass.edu

**Philip S. Thomas**
University of Massachusetts
pthomas@cs.umass.edu

## Abstract

Evaluating policies using off-policy data is crucial for applying reinforcement learning to real-world problems such as healthcare and autonomous driving. Previous methods for *off-policy evaluation* (OPE) generally suffer from high variance or irreducible bias, leading to unacceptably high prediction errors. In this work, we introduce STAR, a framework for OPE that encompasses a broad range of estimators—which include existing OPE methods as special cases—that achieve lower mean squared prediction errors. STAR leverages state abstraction to distill complex, potentially continuous problems into compact, discrete models which we call *abstract reward processes* (ARPs). Predictions from ARPs estimated from off-policy data are provably consistent (asymptotically correct). Rather than proposing a specific estimator, we present a new framework for OPE and empirically demonstrate that estimators within STAR outperform existing methods. The best STAR estimator outperforms baselines in all twelve cases studied, and even the median STAR estimator surpasses the baselines in seven out of the twelve cases.

## 1 Introduction

Within *reinforcement learning* (RL), *off-policy evaluation* (OPE) is the foundational challenge of evaluating the performance, $J(\pi)$, of policies $\pi$ that are different from the ones used to generate data. OPE methods are a general-purpose tool that can be used as part of a local policy search algorithm [45] to provide insight about policies similar to the current policy, or as a tool to evaluate policies without requiring their actual deployment for high-risk applications like those in healthcare [38], education [35, 15], and recommendation systems [5, 7]. Despite many recent advances in OPE, existing methods struggle to give accurate predictions for many real-world applications [52], showing the need for new perspectives on OPE.

Existing methods can be broadly divided into two categories: *importance sampling* (IS) based and model-based [58]. IS-based methods are typically consistent (i.e., their predictions converge probabilistically to the correct value in the limit as the amount of data approaches infinity), but have variance that increases exponentially with the horizon [29, 31]. Model-based methods have lower variance but often introduce bias due to model class mismatch and are not generally guaranteed to be consistent [36, 10]. A third set of methods, which we call *mixture methods*, combine the predictions obtained from both of these categories [22, 53]. However, in some cases, combining the predictions also combines the drawbacks—high variance and bias. This leads us to ask: *Can we develop a framework for OPE that yields predictions that are both consistent and low variance?*

38th Conference on Neural Information Processing Systems (NeurIPS 2024).

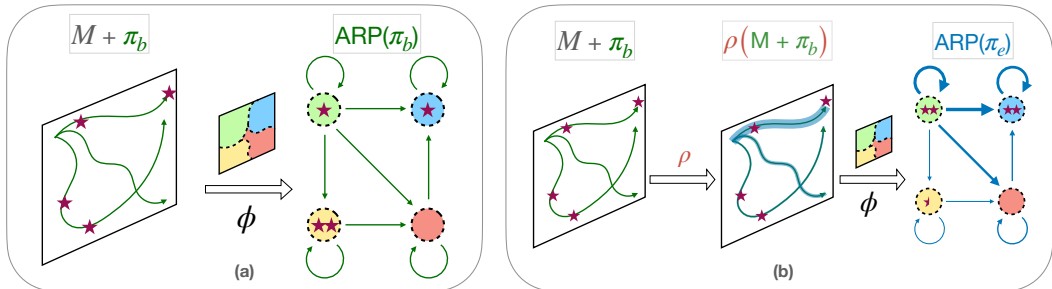

Figure 1: **(a):** MDP $M$ and policy $\pi_b$ are transformed into a discrete *abstract reward process* (ARP) using a state abstraction function $\phi$. The ARP aggregates rewards (denoted by stars) and transition probabilities from all states that map to each abstract state. **(b):** A model of the ARP for the evaluation policy $\pi_e$ is constructed by: reweighting data generated by $\pi_b$ with importance weights $\rho$ (middle), applying the state abstraction function $\phi$, and performing weighted maximum likelihood estimation of the ARP (right). The expected return of a model of this ARP estimated from off-policy data is a *consistent* estimator of the expected return of $\pi_e$.

In this paper, we introduce a new framework that attains this goal by combining the *machinery* underlying IS-based and model-based approaches (not just their *predictions*). Our proposed framework is a fundamentally different approach to OPE that incorporates importance sampling *into* model learning for OPE. Our approach is motivated by the intuition that humans build small mental models of their environment to plan and predict, selectively abstracting away information that is not relevant to the problem at hand [54]. Similarly, complex sequential decision processes can be distilled into compact models that hold sufficient information for (off-)policy evaluation. Specifically, we propose creating small tabular models (even for continuous environments), which we call *abstract reward processes* (ARPs), customized for the problem at hand and for the policy being evaluated. We call this framework for constructing a range of ARPs state-abstract reward processes (STAR).

**Idea Summary:** A *Markov decision process* (MDP) combined with a policy $\pi$ induces a Markov chain with rewards, called a *Markov reward process* (MRP). A model of this process can be estimated to evaluate policy $\pi$. However, two main challenges arise: (1) like any model-based approach, estimating an MRP can introduce asymptotic bias if the chosen model class cannot represent the underlying MRP; and (2) the model of the MRP must be accurately estimated from *off-policy* data, i.e., data generated by a behavior policy $\pi_b$ that differs from the evaluation policy $\pi_e$. The proposed framework addresses both challenges by modeling a special instantiation of an MRP, as detailed next.

First, to address potential model class mismatch, we propose mapping the large and possibly continuous set of states of the MDP to a finite set of *abstract states* using a discrete state abstraction function. We then represent the resulting MRP defined over abstract states, referred to as the *abstract reward process* (ARP), using tabular models. Since the ARP is finite, tabular models can represent it accurately, as depicted in Figure 1(a). However, using a discrete state abstraction may lead to loss of state information that can potentially introduce modeling errors. Surprisingly, we prove that despite state information being abstracted away, the maximum likelihood model of an ARP estimated from *on-policy* data provides consistent estimates of the expected return of the behavior policy $\pi_b$.

Next, to estimate a model of the ARP corresponding to $\pi_e$ from data generated by $\pi_b$, we reweight occurrences of abstract states in the off-policy dataset using importance sampling, see Figure 1(b). In expectation, this has the effect of updating the abstract state visitation counts to reflect those resulting from the policy being evaluated. We prove that the weighted maximum likelihood estimate of a model of the ARP estimated from this off-policy dataset provides consistent estimates of the performance of the evaluation policy.The integration of importance sampling *into* model estimation permits a favorable interpretation of weight clipping for mitigating variance (see Section 4.1).

The STAR framework offers two adjustable knobs—the state abstraction function, and the amount of weight clipping—that instantiate a range of OPE estimators. Varying the configurations of these knobs results in different bias-variance trade-offs for OPE, with existing OPE methods forming special cases of the range of estimators that lie within this framework.

**Contributions:** We empirically evaluate estimators instantiated in this framework on synthetic domains and a healthcare simulator built from real-world ICU data, where the best STAR estimator significantly outperforms baselines in all cases, and even the median STAR estimator surpasses baselines in seven out of twelve cases. It must be emphasized that this work does not propose a specific estimator for OPE; rather, it introduces a fundamentally different framework that offers fresh insights on approaches for off-policy evaluation. These insights open up exciting avenues for new research questions and future directions. In this paper, we introduce:

(1) the first model-based approach for OPE that guarantees asympotic correctness of the estimates without model class assumptions, even for continuous state MDPs (Theorem 4.1).

(2) the concept of *abstract reward processes* for consistent OPE. ARPs abstract away the complexity of the underlying problem, and distill sufficient information for accurate policy evaluation (Theorem 3.1). Being finite, they can be consistently estimated.

(3) a generalizing framework that provides a fresh perspective on OPE by merging the machinery of model-based and IS-based approaches. The framework offers two tunable knobs, various configurations of which instantiate a range of OPE estimators with varying bias-variance characteristics. Existing model-free and model-based methods are special cases in this framework.

## 2 Background and Notation

An MDP is a tuple $M := (\mathcal{S}, \mathcal{A}, p, r, \gamma, \eta)$ where $\mathcal{S}$ is the set of states, $S_t$ is the state at time $t \in \{0, 1, \dots\}$, $\mathcal{A}$ is the set actions, $A_t$ is the action at time $t$, $p : \mathcal{S} \times \mathcal{A} \times \mathcal{S} \rightarrow [0, 1]$ is the *transition function* that characterizes state transition dynamics according to $p(s, a, s') := \Pr(S_{t+1} = s' | S_t = s, A_t = a)$, $r : \mathcal{S} \times \mathcal{A} \rightarrow \mathbb{R}$ is the *reward function* that characterizes rewards according to $r(s, a) := \mathbb{E}[R_t | S_t = s, A_t = a]$, $\gamma \in [0, 1]$ is the reward discount parameter, and $\eta : \mathcal{S} \rightarrow [0, 1]$ characterizes the initial state distribution according to $\eta(s) := \Pr(S_0 = s)$.[1] A policy $\pi : \mathcal{S} \times \mathcal{A} \rightarrow [0, 1]$ characterizes how actions can be selected given the current state according to $\pi(s, a) := \Pr(A_t = a | S_t = s)$. We consider finite horizon MDPs [49] where episodes terminate by some (unspecified) time $T \in \mathbb{N}$—which is common in practical applications of OPE. For simplicity, we set $\gamma = 1$, allowing us to omit $\gamma$ terms.

For OPE, a dataset $\mathcal{D}_n^{(\pi_b)}$ is collected by deploying a behavior policy $\pi_b$ on the MDP $M$. The dataset of $n$ logged trajectories is denoted by $\mathcal{D}_n^{(\pi_b)} := \{H^i\}_{i=1}^n$ where each $H^i := (S_0^i, A_0^i, R_0^i, S_1^i, \dots)$ represents an independent trajectory generated by executing $\pi_b$. The performance of an evaluation policy $\pi_e$ is its expected return, denoted by[2]

$$J(\pi_e) := \mathbb{E}\left[\sum_{t=1}^{T} R_t; \pi_e\right]. \tag{1}$$

The problem of off-policy evaluation entails estimating $J(\pi_e)$ with access only to data $\mathcal{D}_n^{(\pi_b)}$, generated by a behavior policy $\pi_b$, without additional interaction with the MDP. To ensure that samples in $\mathcal{D}_n^{(\pi_b)}$ are sufficiently informative, we make the common assumption that any outcome under $\pi_e$ has non-negligible probability of occurring under $\pi_b$.

**Assumption 2.1.** There exists an (unknown) $\varepsilon > 0$ such that for all $s \in \mathcal{S}$ and $a \in \mathcal{A}$, $(\pi_b(s, a) < \varepsilon) \implies (\pi_e(s, a) = 0)$.

**Background:** For a detailed review of OPE methods, we refer the reader to surveys by Voloshin et al. [58] and Uehara et al. [56]. Concepts fundamental to this approach are briefly introduced here.

1. **Importance Sampling:** Importance sampling [25] enables unbiased estimation of the expected value, $\mathbb{E}[f(X)]$, of a function $f$ applied to a random variable $X \sim p$, given samples of a different random variable $Y \sim q$. The importance sampling estimator is $(p(Y)/q(Y))f(Y)$, where

---

[1]For simplicity, our notation assumes that states, actions, and rewards are discrete random variables allowing for discussion of probabilities, rather than densities or measure theoretic probability. However, the methods proposed in this work extend to MDPs with continuous states, actions, and rewards. Furthermore, although we focus on OPE for MDPs, the method that we propose also applies to *partially observable* MDPs (POMDPs).

[2]We write $; \pi$ within statements of probability or expectations to indicate that random variables like $S_t, A_t$, and $R_t$ result from the use of policy $\pi$.

$p(Y)/q(Y)$ is a term called an importance weight. This technique can provide unbiased estimates (i.e., $\mathbb{E}\left[(p(Y)/q(Y))f(Y)\right] = \mathbb{E}\left[f(X)\right]$) and has proven effective for variance reduction in Monte Carlo sampling [44] and for model-free OPE in RL [42].

2. **State Abstraction:** State abstraction aims to reduce the size of the state space by grouping together similar states in a way that does not change the essence of the underlying problem [28, 43, 1]. A state abstraction function $\phi : \mathcal{S} \to \mathcal{Z}$ lies in the set of functions $\phi \in \Phi$ that map each state $s \in \mathcal{S}$ to an abstract state $z \in \mathcal{Z}$. We consider abstraction functions that partition $\mathcal{S}$ into disjoint sets, where $\mathcal{Z}$ is a finite set.

**Notation:** Indicator functions are abbreviated for clarity. For example, $\mathbf{1}_t^i\{z, z'\} := \mathbf{1}\{\phi(S_{t+1}^{(i)}) = z', \phi(S_t^{(i)}) = z\}$ denotes the occurrence of abstract states $z$ and $z'$ at time steps $t$ and $t+1$ in the $i^{\text{th}}$ logged trajectory, with $\mathbf{1}_t^i\{z\}$ defined correspondingly. The expected return of $\pi$ when obtained from $O$—where $O$ may be the MDP, or an ARP—is denoted by $J(\pi; O)$. The sample estimate of a variable $y$ estimated from $n$ samples is denoted by $\hat{y}_n$. Summation limits are often dropped for brevity, with $\sum_t$ denoting $\sum_{t=0}^T$ and $\sum_i$ denoting $\sum_{i=1}^n$.

### 2.1 Markov Reward Processes

A *Markov reward process* (MRP) extends the idea of a Markov chain by associating states with rewards. Formally, an MRP is a tuple $(\mathcal{X}, p, r, \gamma, \eta)$ where $\mathcal{X}$ is the set of states of the MRP, $X_t$ is the state at time $t$, $p : \mathcal{X} \times \mathcal{X} \to [0, 1]$ is the transition function where $p(x, x') := \Pr(X_{t+1}=x'|X_t=x)$, $r : \mathcal{X} \to \mathbb{R}$ is the reward function where $r(x) := \mathbb{E}[R_t|X_t=x]$, $\gamma \in [0, 1]$ is the discount factor, and $\eta : \mathcal{X} \to [0, 1]$ is the starting state distribution. We consider finite horizon MRPs where episodes terminate by some (unspecified) timestep and set $\gamma = 1$.

A specific MRP is induced by the use of a fixed policy $\pi$ on an MDP $M$, where $\mathcal{X} = \mathcal{S}$. The resulting transition and reward functions, denoted by $p^\pi$ and $r^\pi$ respectively, are:

$$p^\pi(x, x') = \frac{\sum_t \Pr(S_{t+1} = x', S_t = x; \pi)}{\sum_t \Pr(S_t = x; \pi)}, \quad r^\pi(x) = \frac{\sum_t \mathbb{E}\left[R_t|S_t = x; \pi\right] \Pr(S_t = x; \pi)}{\sum_t \Pr(S_t = x; \pi)}. \quad (2)$$

The Markov property [37] allows for further simplification of the above expressions (detailed in Appendix A.1), but this form is most conducive to our subsequent discussion. In this work, we focus on a specific instantiation of an MRP, described in the next section, where the set of states $\mathcal{X}$ of the MRP are outputs of a state abstraction function $\phi \in \Phi$.

## 3 Abstract Reward Processes

An abstract reward process is a Markov reward process—derived from MDP $M$ and policy $\pi$ and defined over *abstract* states—that we use to evaluate $\pi$. The ARP provides two primary benefits for policy evaluation: (1) it preserves sufficient information to exactly evaluate the policy $\pi$, and (2) the ARP can be *consistently* estimated from data. In this section, we formalize the concept of an ARP, and highlight the theoretical and practical benefits of using ARPs for policy evaluation.

Given a state abstraction function $\phi : \mathcal{S} \to \mathcal{Z}$, the ARP $\mathfrak{R}_\phi^\pi$ is defined such that $\mathcal{X} = \mathcal{Z}$. Formally, $\mathfrak{R}_\phi^\pi$ is an MRP $(\mathcal{Z}, P_\phi^\pi, R_\phi^\pi, \eta_\phi)$, with $\gamma = 1$ (see Appendix A.1 for a discussion on termination in ARPs and MRPs). The components of the ARP are defined over *abstract* states as:

$$P_\phi^\pi(z, z') := \frac{\sum_t \Pr(\phi(S_{t+1})=z', \phi(S_t)=z; \pi)}{\sum_t \Pr(\phi(S_t)=z; \pi)}, R_\phi^\pi(z) := \frac{\sum_t \mathbb{E}[R_t|\phi(S_t)=z; \pi] \Pr(\phi(S_t)=z; \pi)}{\sum_t \Pr(\phi(S_t)=z; \pi)}, \quad (3)$$

and $\eta_\phi(z) := \Pr(\phi(S_0) = z)$. These expressions *cannot* be simplified further, unlike the case of an MRP [2]. Since $\mathcal{Z}$ is a finite set, i.e., the abstract states are discrete, the components of the ARP can be represented by matrices (we use uppercase letters to emphasize this). The expected return of $\mathfrak{R}_\phi^\pi$ can be computed efficiently using a linear solver to evaluate the expression $J(\pi; \mathfrak{R}_\phi^\pi) := (\mathrm{I} - P_\phi^\pi)^{-1} R_\phi^\pi \eta_\phi$, or via Monte Carlo rollouts of the reward process.

**ARPs are *Performance Preserving*:** The expected return of an ARP has a surprising property: even though some state information is abstracted away to create simple discrete abstract states, the finite

ARP, derived from a possibly continuous and complex MDP, preserves sufficient information about the performance of the policy that defines the ARP *for all* $\phi \in \Phi$.

**Theorem 3.1.** $\forall \phi \in \Phi$, *the performance of a policy $\pi$ is equal to the expected return of the abstract reward process $\mathfrak{R}_\phi^\pi$ defined from MDP $M$, i.e., $J(\pi; \mathfrak{R}_\phi^\pi) = J(\pi; M)$.*

*Proof.* See Appendix B.1. □

The result holds for the ground-truth ARP $\mathfrak{R}_\phi^\pi$. In practice, a model of the ARP must be estimated from data. Next, we describe how the choice of defining an ARP over discrete abstract states eliminates model class mismatch, enabling asymptotically correct estimation of the ARP from data.

**Eliminating Model Class Mismatch:** Methods that learn models from data make an assumption about the class of models used to represent the data. A significant challenge is that of *model class mismatch*, where this assumed model class is often unable to represent the true data distribution. As an example, a neural network parameterizing a univariate Gaussian distribution cannot accurately represent data generated from a bimodal distribution. In the context of this work, the transition function of an ARP may specify arbitrary probability distributions over discrete abstract states, necessitating a careful selection of the model class. *Tabular models* are capable of representing *any* distribution over discrete variables. Therefore, using tabular models when estimating an ARP from data ensures that there is no model class mismatch. This is why we employ state abstraction functions $\phi \in \Phi$ that partition the state space into a finite number of disjoint sets, or discrete abstract states. The abstraction functions may be viewed as: (a) a discrete clustering of the state space, or (b) a discretization of continuous states.

While this addresses model class mismatch, the use of a discrete state abstraction may itself be a source of modeling error. Mapping groups of (possibly continuous) states to discrete abstract states loses information about the state of the MDP. A process defined over the abstract states cannot in general capture the full complexity of the underlying MDP and policy. Nonetheless, Theorem 3.1 guarantees that the ARP is performance-preserving, ensuring that the use of discrete state abstractions is not a source of error for policy evaluation. Additionally, since we have eliminated model class mismatch, a perfect model of the ARP can be asymptotically estimated.

To estimate the ARP from $\mathcal{D}_n^{(\pi_b)}$, apply the state abstraction function to states in $\mathcal{D}_n^{(\pi)}$ to map them to the abstract state space. Denote the *maximum likelihood estimate* of the model of the ARP obtained from the dataset (with abstract states) by $\widehat{\mathfrak{R}}_{n,\phi}^\pi := (\mathcal{Z}, \widehat{P}_{n,\phi}^\pi, \widehat{R}_{n,\phi}^\pi, \widehat{\eta}_{n,\phi})$. The components take the form:

$$\widehat{P}_{n,\phi}^\pi(z, z') = \frac{\sum_{i,t} \mathbf{1}_t^i\{z, z'\}}{\sum_{i,t} \mathbf{1}_t^i\{z\}}; \quad \widehat{R}_{n,\phi}^\pi(z) = \frac{\sum_{i,t} \mathbf{1}_t^i\{z\} R_t^i}{\sum_{i,t} \mathbf{1}_t^i\{z\}}; \quad \widehat{\eta}_{n,\phi}^\pi(z) = \frac{\sum_{i=1}^n \mathbf{1}^i\{z_0 = z\}}{n} \quad (4)$$

**Asymptotic Correctness:** With access to *on-policy data* $\mathcal{D}_n^{(\pi)}$, the following result states that ARPs enable consistent model-based estimation of the policy's performance.

**Lemma 3.2.** $\forall \phi \in \Phi$, *the expected return of the maximum likelihood estimate $\widehat{\mathfrak{R}}_{n,\phi}^\pi$ converges almost surely to the expected return of the policy $\pi$, i.e., $J(\pi; \widehat{\mathfrak{R}}_{n,\phi}^\pi) \xrightarrow{a.s.} J(\pi; M)$.*

*Proof.* See Appendix B.2. □

As the amount of data ($n$) increases, the estimate of $J(\pi; M)$ becomes increasingly accurate, i.e., the return estimate is consistent. This result holds for all $\phi \in \Phi$. It implies that *even an arbitrarily small model derived from a large, complex sequential decision-making problem will not introduce asymptotic bias.* However, this theoretical guarantee requires on-policy data and so does not directly assist us in off-policy evaluation. To that end, we introduce a procedure for estimation of the ARP from *off-policy data* that merges the machinery of IS-based and model-based methods.

### 3.1 Estimation from Off-Policy Data: Weighted Maximum Likelihood Estimation

We present a method for consistent estimation of the ARP corresponding to the evaluation policy $\pi_e$ from off-policy data $\mathcal{D}_n^{(\pi_b)}$. It relies on the following intuition:

> The expected value of the indicator function of an event represents the probability of that event. Use importance sampling to approximate the probability of that event under a different distribution.

To estimate an ARP from off-policy data, assign importance weights $\rho_{0:t}$ to the abstract states $(Z_t := \phi(S_t))$ in the dataset $\mathcal{D}_n^{(\pi_b)}$. Let $H_t := (S_0, A_0, R_0, \ldots, S_{t-1}, A_{t-1}, R_{t-1}, S_t, A_t)$ denote a sub-trajectory up to time $t$. The importance weight $\rho_{0:t}$ is then the ratio of the probability of $H_t$ under $\pi_e$ and $\pi_b$, i.e., $\rho_{0:t} := \frac{\Pr(H_t; \pi_e)}{\Pr(H_t; \pi_b)} = \prod_{j=0}^{t} \frac{\pi_e(S_j, A_j)}{\pi_b(S_j, A_j)}$.[3] The maximum likelihood estimate (MLE) of $\mathfrak{R}_\phi^{\pi_e}$ obtained from the weighted off-policy data is denoted by $\widehat{\mathfrak{R}}_{n,\phi}^{\pi_b \to \pi_e} :=$ $\left( \mathcal{Z}, \widehat{P}_{n,\phi}^{\pi_b \to \pi_e}, \widehat{R}_{n,\phi}^{\pi_b \to \pi_e}, \widehat{\eta}_{n,\phi}^{\pi_b \to \pi_e} \right)$, where $\widehat{\eta}_{n,\phi}^{\pi_b \to \pi_e}(z) = \frac{\sum_{i=1}^{n} \mathbf{1}^i \{z_0 = z\}}{n}$ remains unchanged, and

$$\widehat{P}_{n,\phi}^{\pi_b \to \pi_e}(z, z') = \frac{\sum_{i,t} \mathbf{1}_t^i \{z, z'\} \rho_{0:t}}{\sum_{i,t} \mathbf{1}_t^i \{z\} \rho_{0:t}}, \qquad \widehat{R}_{n,\phi}^{\pi_b \to \pi_e}(z) = \frac{\sum_{i,t} \mathbf{1}_t^i \{z\} \rho_{0:t} R_t^i}{\sum_{i,t} \mathbf{1}_t^i \{z\} \rho_{0:t}}. \tag{5}$$

This estimation is a form of weighted maximum likelihood estimation [13]. Including the importance ratios in the numerator and denominator of the estimated transition and reward functions of the ARP enables estimation from off-policy data generated by $\pi_b$. The estimated model of the ARP is consistent and, as shown next, allows for consistent off-policy evaluation.

**Lemma 3.3.** *Under Assumption 2.1, the weighted maximum likelihood estimate $\widehat{\mathfrak{R}}_{n,\phi}^{\pi_b \to \pi_e}$ converges almost surely to the ground-truth ARP $\mathfrak{R}_\phi^{\pi_e}$, i.e., $\widehat{\mathfrak{R}}_{n,\phi}^{\pi_b \to \pi_e} \xrightarrow{a.s.} \mathfrak{R}_\phi^{\pi_e}$.*

*Proof.* See Appendix B.3. $\qquad\qquad\qquad\qquad\qquad\qquad\qquad\qquad\qquad\qquad\qquad\qquad\qquad\quad$ □

## 4 Off-Policy Evaluation with ARPs

The expected return of $\widehat{\mathfrak{R}}_{n,\phi}^{\pi_b \to \pi_e}$ is a consistent estimate of the performance of policy $\pi_e$ since $\widehat{\mathfrak{R}}_{n,\phi}^{\pi_b \to \pi_e}$ is an asymptotically correct estimate of $\mathfrak{R}_\phi^{\pi_e}$, as per Lemma 3.3.

**Theorem 4.1.** *The expected return of the ARP $\widehat{\mathfrak{R}}_{n,\phi}^{\pi_b \to \pi_e}$ (built from off-policy data) converges almost surely to the expected return of $\pi_e$, i.e., $J(\pi_e; \widehat{\mathfrak{R}}_{n,\phi}^{\pi_b \to \pi_e}) \xrightarrow{a.s.} J(\pi_e; M)$.*

*Proof.* See Appendix B.4. $\qquad\qquad\qquad\qquad\qquad\qquad\qquad\qquad\qquad\qquad\qquad\qquad\qquad\quad$ □

To our knowledge, this is the *first* instance of a model-based OPE method that comes with the theoretical guarantee of consistent performance estimates, even for continuous problems and without model class assumptions. So far, we have achieved one of the starting goals—that of *consistency*. However, the use of importance weights $\rho_{0:t}$ for weighted MLE is expected to introduce high variance. Next, we discuss methods to mitigate the variance of IS.

### 4.1 Variance Reduction: Leveraging *Markovness* of the State Abstraction

A common technique for mitigating the variance of IS-based methods for OPE is clipping the importance weights to the $c$ most recent ratios [18, 3, 20], i.e., $\rho_{(t-c+1)^+:t} := \prod_{i=(t-c+1)^+}^{t} \frac{\pi_e(S_i, A_i)}{\pi_b(S_i, A_i)}$, where $(t - c + 1)^+ := \max(t - c + 1, 0)$. This is often a bad approximation for classical IS-based methods, as it implies that only the $c$ most recent actions affect the reward distribution at any timestep, which rarely holds true in practice. In STAR, importance weights are incorporated into model estimation, resulting in a more reasonable implication of weight clipping.

By importance weighting the abstract-state occurrences as described in Equation (5), clipping importance weights, in this case, implies *that the $c$ most recent abstract states are sufficient to determine the current abstract-state transition and reward distributions*. This allows actions from the distant past to influence the current reward, as *the effects of actions propagate through the abstract state transitions*, unlike in IS-based methods. This condition, that a recent history of abstract states is sufficient to predict the current abstract state transition distribution, often approximately holds

---

[3]The expression for the importance weight can be extended to continuous and hybrid probability measures using Radon-Nikodym derivatives. As with other terms, we hereafter denote the importance weight for the $i$[th] episode as $\rho_{0:t}^i$.

in practice as discussed in POMDP literature [30, 39]. While the approximation may introduce asymptotic bias in exchange for reduced variance, certain abstraction functions that satisfy specific conditions can incur no asymptotic bias.

**Weight Clipping without Asymptotic Bias:** Intuitively, the use of $c$-clipped importance weights, $\rho_{(t-c+1)^+:t}$, updates the estimated distribution of the previous $c$ abstract states—as if under the evaluation policy—while leaving the ones before unchanged. $c$-clipping does not introduce asymptotic bias when the previous $c$ abstract states form a sufficient statistic for predicting the current abstract state transition distribution. This notion of conditional independence from history given the recent past is referred to as the *Markovness* of the abstraction function $\phi$. We posit that there exist abstraction functions that are $c$-th order Markov [51, 9].

**Definition 4.2** ($c$-th order Markov). The abstraction function $\phi$ is $c$-th order Markov if $\Pr(\phi(S_{t+1})|\phi(S_t), \cdots, \phi(S_{(t-c+1)^+}); \pi) = \Pr(\phi(S_{t+1})|\phi(S_t), \cdots, \phi(S_0); \pi)$ for $\pi \in \{\pi_b, \pi_e\}$.

Let $\widehat{\mathfrak{R}}_{\phi,c}^{\pi_b \to \pi_e}$ denote the ARP estimated using $c$-clipped importance weights, $\rho_{(t-c+1)^+:t}$, in place of $\rho_{0:t}$ in Equation (5).

**Theorem 4.3.** *Given a $c$-th order Markov $\phi$, the expected return of the abstract reward process $\widehat{\mathfrak{R}}_{\phi,c}^{\pi_b \to \pi_e}$ converges almost surely to the expected return of $\pi_e$, i.e., $J(\pi_e; \widehat{\mathfrak{R}}_{\phi,c}^{\pi_b \to \pi_e}) \xrightarrow{a.s.} J(\pi_e; M)$.*

*Proof.* See Appendix B.5. □

Even when $\phi$ does not satisfy the above condition, weight clipping proves to be a practical approximation and results in low mean squared prediction error, as demonstrated empirically in Section 5. The steps for performing off-policy evaluation by estimating $\widehat{\mathfrak{R}}_{\phi,c}^{\pi_b \to \pi_e}$ are highlighted in Algorithm 1.

## 4.2 Fantastic $\phi$'s and Where to Find Them

Discrete abstraction functions that are $c$-th order Markov with small values of $c$ represent the most suitable abstractions for enabling asymptotically correct, low-variance off-policy evaluation using STAR. An automated approach to discovering such abstraction functions, however, remains elusive. In a manner reminiscent of the options framework [50], wherein one might consider the usefulness of options before having methods for constructing options automatically, this work emphasizes the remarkable effectiveness of state abstractions used in abstract reward processes for OPE. It motivates a research area akin to option discovery: *abstraction discovery for OPE*.

We expect the following factors to play an important role in the search for good abstraction functions: (a) state-visitation distributions of $\pi_b$ and $\pi_e$, determining the granularity of abstraction in different parts of the state set, and (b) the distribution shift in abstract state visitation induced by the two policies, determining the extent of weight clipping that can be applied. Both of these are affected by properties of the underlying MDP, in particular the transition function, and in our initial analyses, we observe varying effects of similar abstractions across different MDPs (Appendix C.2).

We observe that a simple approach of randomly initializing centroids and applying $k$-means clustering [34, 33], where each cluster denotes a discrete abstract state, results in abstractions that provide competitive OPE performance, often significantly outperforming existing methods. We call this naive clustering-based abstraction method CluSTAR, and use it for our experiments. In some cases, abstraction by aggregation of states can increase the difficulty of estimation of the transition function. For example, aggregation of two states with deterministic transitions—which can be estimated perfectly from a single observation of those transitions—creates stochastic transitions between abstract states. However, in general, state aggregation tends to simplify estimation by increasing the effective sample size [21].

**Recovering Existing OPE Methods from STAR:** Different configurations of $(\phi, c)$ induce different ARPs, $\widehat{\mathfrak{R}}_{\phi,c}^{\pi_b \to \pi_e}$. For certain configurations of $(\phi, c)$:

- $|\mathcal{Z}| = 1$ and no weight clipping: Mapping all states to a single abstract state yields the *weighted per-decision importance sampling* (WPDIS) estimator [42].
- $\mathcal{Z} = \mathcal{S}$ and $c = 1$: Amounts to no state abstraction, and yields the maximum likelihood estimate of the MRP over states. The MRP is a combination of the approximate-model estimator [40] that directly estimates the model dynamics with the evaluation policy.

The recovery of these familiar estimators at the endpoints of STAR highlights the unifying nature of the framework. More importantly, the intermediate configurations of $(\phi, c)$ uncover a whole new set of OPE estimators. The ARPs in this space often inherit a mixture of the favorable properties of both of the endpoints. Consequently, the framework yields estimators that can significantly outperform existing methods, as shown empirically in the next section.

## 5 Empirical Analysis

In this section, we **(A)** analyze the performance of the set of estimators (ARPs) induced by STAR across different configurations of $(\phi, c)$, and **(B)** compare the performance of the best and median ARPs from this set against existing OPE methods to demonstrate that estimators encompassed by STAR often outperform prior OPE methods. We use the following domains for OPE: **(1)** CartPole [49]: A classic control domain in OpenAI Gym [6]. **(2)** ICU-Sepsis [8]: An MDP that simulates treatment of sepsis in the ICU. ICU-Sepsis is built from real-world medical records obtained from the MIMIC-III dataset [24], using a modified version of the process described by Komorowski et al. [26]. **(3)** Asterix from the MinAtar testbed [61]: A miniaturized version of the Atari game Asterix. Details about each domain, and about the behavior and evaluation policies can be found in the Appendix C. The code is available at: `https://github.com/shreyasc-13/STAR`.

**(A) Estimator Selection:** Estimator selection presents a significant challenge for OPE [48] due to the unavailability of a *validation set*. To select STAR estimators to compare against existing methods, we first report the performance of the set of the ARPs induced by STAR, across a range of configurations of $(\phi, c)$. We highlight the performance of the best and median estimators from this set. For reference, we compare the mean squared prediction errors from the estimated ARPs against WPDIS and approximate-model estimator (MBased), the two endpoints of STAR, as shown in Figure 2 on the CartPole domain. The state abstraction is performed with CluSTAR with the number of centroids $|\mathcal{Z}| \in \{2, 4, 8, 16, 32, 64, 128\}$ and the weight clipping factor $c \in \{1, 2, 3, 4, 5\}$ defining 35 ARPs, where the performance of each is indicated by ●. This range of

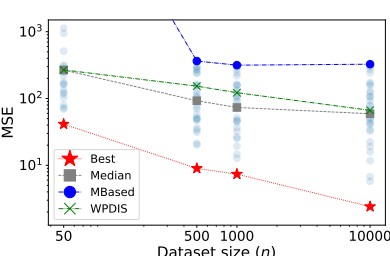

Figure 2: Mean squared prediction errors of the estimated ARPs for the set of hyperparameters swept over for Cart-Pole.

$(\phi, c)$ is picked based on the intuition that (a) larger values of $c$ are expected to introduce high variance, and (b) this range of $|\mathcal{Z}|$ covers a variety of granularities of state abstraction (for a continuous problem). The results are averages across 200 trails. The prediction errors for the estimated ARPs are competitive with baselines. This suggests that *an average estimator in STAR is competitive with existing OPE methods*, and even better performance may be attained by using specialized methods for abstraction discovery and estimator selection [55]. Figure 4, in the Appendix, provides a detailed breakdown of the performance of each ARP in the set considered, presented as a heatmap. This highlights patterns observed for varying values of $|\mathcal{Z}|$ and $c$ across different domains. Further discussion and details about the best-performing configurations of $(\phi, c)$ are deferred to Appendix C.2.

**(B) ARPs Can Outperform Existing Methods for OPE:** We compare against representative methods from the main categories of OPE methods: (a) IS-based methods: Vanilla IS, Per-Decision IS, Weighted IS, Weighted Per-Decision IS [42]; (b) Model-based methods: that approximate a model of the MDP—MBased [49]—and directly estimate off-policy Q-values—FQE and MRDR [27, 11]; and (c) Mixture methods: DR [22] and MAGIC [53], which blend estimates from the methods in the aforementioned categories. We use implementations from the Caltech OPE Benchmarking Suite (COBS) [58] for all methods. The representative method for minmax style estimators, IH [31], is designed for the infinite horizon setting and performs poorly with $\gamma = 1$ [63]. Due to the instability of IH estimates in our experiments, we excluded it from comparison.

**Results [Figure 3 and 4]:** The ranges of $|\mathcal{Z}|$ and $c$ for each domain, which induces a set of ARPs that we consider, are detailed in Appendix C.2. Figure 4 in Appendix C.2 demonstrates the performance of each ARP from that set as a heatmap, showing that the best configuration of $(\phi, c)$ varies with the dataset size for each domain. In critical applications like sepsis treatment, as simulated in ICU-Sepsis, where incorrect policy performance estimates can be catastrophic, the best estimators in

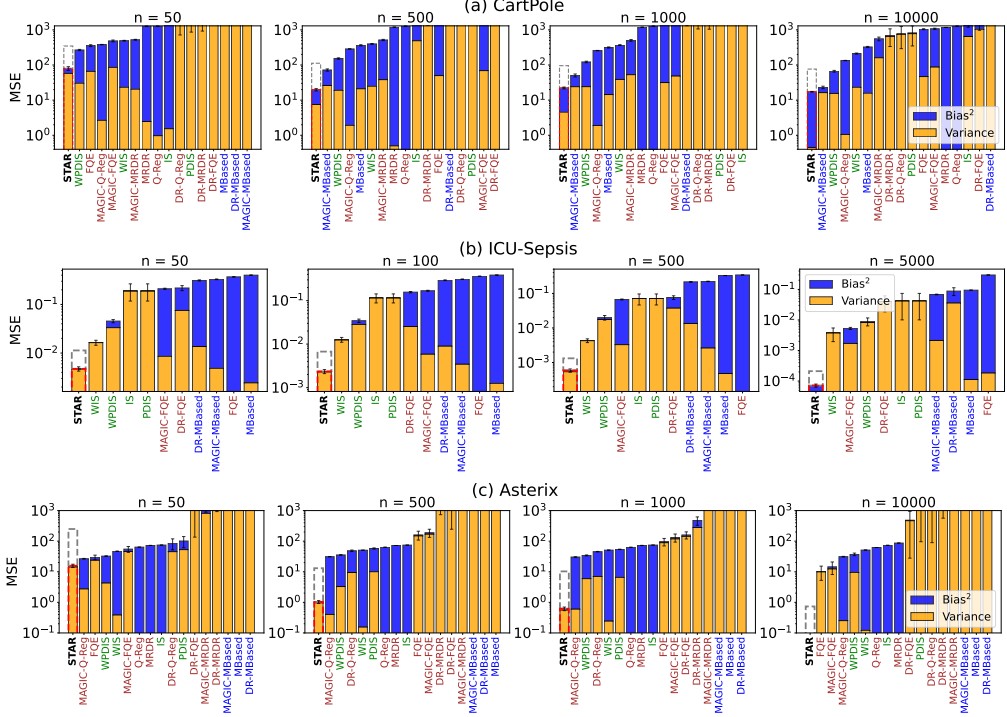

Figure 3: Mean squared prediction errors of best and median ARPs from STAR compared against existing OPE methods. The empirically estimated bias-variance decomposition of the error is shown. The results are averaged over 200 trials, with error bars indicating standard error. Note: For ICU-Sepsis, regression-based methods (MRDR and Q-Reg) were computationally intractable due to the large state set, as the corresponding Weighted Least Squares methods for regression were too slow. In all domains and across all datasizes, the best ARP in STAR outperforms baselines in all cases, and the even the median estimator does so in 7 out of 12 cases.

STAR achieve prediction errors that are an order of magnitude lower than the best baseline. These results in Figure 3 emphasize that *highly compact ARPs that distill complex sequential processes are particularly effective for off-policy evaluation*. The best ARP in STAR for each domain abstracts: (a) the continuous state space of CartPole to $|\mathcal{Z}| = 32$ abstract states, (b) the 747 states of ICU-Sepsis to $|\mathcal{Z}| = 16$ abstract states, and (c) the 400 states of Asterix to $|\mathcal{Z}| = 8$ abstract states, to construct compact finite ARPs for OPE. Furthermore, the horizon length of CartPole is 50, and episodes in ICU-Sepsis and Asterix go up to 120 and 75 timesteps respectively. Such relatively long horizons have proven to be challenging for prior OPE methods. STAR leverages ARPs to enable OPE at scales commonly seen in practice.

## 6 Related Work

The problem of off-policy evaluation (OPE) has been extensively studied due to its relevance for practical applications of reinforcement learning [38, 35]. Extensive surveys on the topic, both theoretical [56] and empirical [58, 14] delineate the numerous approaches to the problem. Model-based approaches for OPE have proven effective [32, 62] but are restricted by the model class used. In this work, we use state abstraction to define compact models called abstract reward processes, and demonstrate their effectiveness as off-policy estimators. Historically, state abstraction research has focused on grouping similar states in a way that does not change the essence of the underlying problem [28, 43, 1], to reduce the complexity of the problem. However, the use of state abstractions for OPE remains under-explored. Pavse and Hanna [41] show that the use of state abstraction with marginalized importance sampling achieves variance reduction in high-dimensional state spaces, but their approach does not use abstraction to construct models. Jiang et al. [23] study abstraction selection for model-based RL, balancing model complexity and policy value suboptimality. Abstraction discovery or

learning has focussed on discretization of continuous state spaces to reduce problem complexity [47], and distilling the Markov features [2] or reward relevant features [12, 59].

## 7 Discussion and Conclusion

In this paper, we have introduced a new framework for consistent model-based off-policy evaluation. This framework leverages state abstraction to prevent model class mismatch along with importance sampling to consistently learn models from off-policy data. Unlike traditional model-based methods, our approach eliminates the need for model class assumptions and provides theoretical guarantees for the obtained performance estimates. Moreover, using state abstraction increases the effective sample size [21], which is particularly beneficial in limited data regimes. Importantly, this work presents a framework with a new approach to OPE, rather than a specific new method. Estimators that lie within this framework significantly outperform existing OPE methods, with the best estimator consistently outperforming all baselines, as demonstrated in our empirical evaluation.

The framework has two main limitations: it requires knowledge of the probabilities of observed actions under the behavior policy, which may not always be available, and a principled method for selecting well-performing configurations of the abstraction function and the weight clipping factor remain elusive. Combining this work with regression IS [19] would be a practical extension that addresses the first limitation. Additionally, a data-driven approach to automated estimator selection based on characteristics of the domain, dataset sizes, and other factors, as suggested by Su et al. [48], would enhance its practical application.

Our findings indicate that even a simple class of abstraction functions can provide competitive OPE performance. We theoretically demonstrate the existence of certain abstraction functions that may offer better performance. Investigating the properties of abstraction functions and developing automated approaches to *abstraction discovery* for ARPs are promising directions for future work on creating high-performing OPE methods.

**Acknowledgements** We thank Yash Chandak, Mohammad Ghavamzadeh and Dhawal Gupta for their helpful discussions and feedback on this work. We would also like to thank Josiah Hanna, Bo Liu, Cameron Allen, and the anonymous reviewers for their feedback.

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

# A Preliminaries

**Consistency and Almost Sure Convergence**  Let $\hat{\theta}_n$ denote the estimator of a statistic $\theta$, estimated from $n$ data points. As the amount of data approaches infinity, i.e., as $n \to \infty$, the estimator is said to *converge almost surely* to $\theta$ if and only if

$$\Pr\left(\lim_{n \to \infty} \hat{\theta}_n = \theta\right) = 1.$$

We write $\theta_n \xrightarrow{\text{a.s.}} \theta$ to denote that the sequence $\theta_n$ converges almost surely to $\theta$. An estimator that converges almost surely to the true value of the statistic is said to be (strongly) **consistent**.

**Continuous Mapping Theorem**  The continuous mapping theorem [4, 57] states that if a sequence of random variables $(X_i)_{i=1}^n$ converges almost surely to a random variable $X$, a function $f$ has discontinuity points $D_f$, and $\Pr(X \in D_f) = 0$, then the sequence $(f(X_i))_{i=1}^n$ converges almost surely to $f(X)$.

**Tower Rule**  For random variables $X$ and $Y$ defined on the same probability space, the tower rule states that the expected value of the conditional expected value of $X$ given $Y$ is the same as the expected value of $X$, i.e.,

$$\mathbb{E}[X] = \mathbb{E}[\mathbb{E}[X \mid Y]].$$

## A.1 Additional Notes on MRPs and ARPs

Here we provide additional details regarding the definitions of MDPs and MRPs. Specifically, we show how the expression of the transition and reward functions of the MRP can be simplified when $\mathcal{X} = \mathcal{S}$ and discuss how to handle termination in ARPs that are derived from finite-horizon MDPs.

### A.1.1 Simplification of Equation 2

The expressions for the components of a Markov reward process (MRP) can be simplified by using the Markov property. The transition function simplifies as:

$$
\begin{aligned}
p^\pi(x, x') &= \frac{\sum_t \Pr(S_{t+1} = x', S_t = x; \pi)}{\sum_t \Pr(S_t = x; \pi)} \\
&= \frac{\sum_t \Pr(S_{t+1} = x' \mid S_t = x; \pi)\Pr(S_t = x; \pi)}{\sum_t \Pr(S_t = x; \pi)} \\
&= \frac{\sum_t \sum_{a \in \mathcal{A}} \Pr(S_{t+1} = x' \mid S_t = x, A_t = a; \pi)\Pr(A_t = a|S_t = x)\Pr(S_t = x; \pi)}{\sum_t \Pr(S_t = x; \pi)} \\
&= \frac{\sum_t \sum_{a \in \mathcal{A}} p(x, a, x')\pi(x, a)\Pr(S_t = x; \pi)}{\sum_t \Pr(S_t = x; \pi)} \\
&= \frac{\sum_{a \in \mathcal{A}} p(x, a, x')\pi(x, a)\left(\sum_t \Pr(S_t = x; \pi)\right)}{\sum_t \Pr(S_t = x; \pi)} \\
&= \sum_{a \in \mathcal{A}} p(x, a, x')\pi(x, a).
\end{aligned}
$$

Similarly, the reward function simplifies as:

$$
\begin{aligned}
r^\pi(x) &= \frac{\sum_t \mathbb{E}[R_t|S_t = x; \pi]\Pr(S_t = x; \pi)}{\sum_t \Pr(S_t = x; \pi)} \\
&= \frac{\sum_t \sum_{a \in \mathcal{A}} r(x, a)\pi(x, a)\Pr(S_t = x; \pi)}{\sum_t \Pr(S_t = x; \pi)} \\
&= \frac{\sum_{a \in \mathcal{A}} r(x, a)\pi(x, a)\left(\sum_t \Pr(S_t = x; \pi)\right)}{\sum_t \Pr(S_t = x; \pi)} \\
&= \sum_{a \in \mathcal{A}} r(x, a)\pi(x, a).
\end{aligned}
$$

### A.1.2 Handling termination in an MDPs, MRPs, and ARPs

Each episode in a finite-horizon MDP concludes by some timestep $T \in \mathbb{N}$, referred to as the *termination* of that episode. In practice, there are two common ways of modeling termination of episodes in finite-horizon MDPs: 1) including an absorbing state $s_\infty$ in the state set, or 2) introducing a termination function $\beta : \mathcal{S} \to [0, 1]$ representing the probability of termination of an episode from each state, i.e., $\beta(s) \coloneqq \Pr(\text{terminate} | S_t = s)$. In the first case, by timestep $T$ the process transitions into the absorbing $s_\infty$ after which it continually transitions back to $s_\infty$ getting a reward of zero. In the second case, after timestep $T$ the process stops and there are no subsequent samples. Expressions involving sums over samples in an epsiode are denoted correspondingly, for example, the expected return $J(\pi) \coloneqq \mathbb{E}[\sum_{t=1}^\infty R_t; \pi]$ in the first case, and $J(\pi) \coloneqq \mathbb{E}[\sum_{t=1}^T R_t; \pi]$ in the second. Both approaches are mathematically equivalent and correspondingly, MRPs induced by the combination of finite-horizon MDPs with a policy can model termination in either way.

However, termination in ARPs requires special attention. The introduction of an absorbing abstract state $z_\infty$ places a condition on the abstraction function $\phi$—it requires the abstraction function to map *only* $s_\infty$, and no other state, to $z_\infty$. In this work, we *implement* termination in code by estimating abstract termination functions that represent the probability of termination from each abstract state. Denote the abstract termination function of an ARP by $\beta_\phi^\pi : \mathcal{Z} \to [0, 1]$. It must be noted that both approaches continue to be mathematically equivalent. The probability of transitioning from any abstract state $z$ to $z_\infty$ is the same as the the probability of termination from that state, i.e., $\beta_\phi^\pi(z) = \mathrm{P}_\phi^\pi(z, z_\infty)$. Note that

$$\mathrm{P}_\phi^\pi(z_\infty, z) = \begin{cases} 0 & \text{if } z \neq z_\infty \\ 1 & \text{if } z = z_\infty. \end{cases}$$

Similarly, the reward function is zero for the absorbing abstract state: $\mathrm{R}_\phi^\pi(z_\infty) = 0$. It is mathematically more succint to model termination with $z_\infty$, and we use this (equivalent) approach in our theoretical analysis.

In our implementation, $z_\infty$ is not defined and thus the components of $\mathfrak{R}_\phi$ do not take $z_\infty$ as input. The equivalant form of the expected return of an ARP, that models a termination function, is then given by:

$$\begin{aligned} J(\pi; \mathfrak{R}_\phi^\pi) &= \mathrm{R}_\phi^\pi \eta_\phi + \left( I - \mathrm{diag}(\beta_\phi^\pi) \right) \left[ \mathrm{P}_\phi^\pi \mathrm{R}_\phi^\pi \eta_\phi + \left( I - \mathrm{diag}(\beta_\phi^\pi) \right) \left[ (\mathrm{P}_\phi^\pi)^2 \mathrm{R}_\phi^\pi \eta_\phi + \ldots \right] \right] \\ &= \mathrm{R}_\phi^\pi \eta_\phi + \left( I - \mathrm{diag}(\beta_\phi^\pi) \right) \mathrm{P}_\phi^\pi \mathrm{R}_\phi^\pi \eta_\phi + \left( I - \mathrm{diag}(\beta_\phi^\pi) \right)^2 (\mathrm{P}_\phi^\pi)^2 \mathrm{R}_\phi^\pi \eta_\phi + \ldots \\ &= \sum_{k=0}^\infty \left( \left( I - \mathrm{diag}(\beta_\phi^\pi) \right) \mathrm{P}_\phi^\pi \right)^k \mathrm{R}_\phi^\pi \eta_\phi. \end{aligned}$$

Correspondingly, the closed form expression for the expected return that accounts for the termination function is given by,

$$J(\pi; \mathfrak{R}_\phi^\pi) = \left( I - \left( I - \mathrm{diag}(\beta_\phi^\pi) \right) \mathrm{P}_\phi^\pi \right)^{-1} \mathrm{R}_\phi^\pi(z) \eta_\phi. \tag{6}$$

Note that $\left( I - \left( I - \mathrm{diag}(\beta_\phi^\pi) \right) \mathrm{P}_\phi^\pi \right)$ is an invertible matrix, as the left hand side of the expression, $J(\pi; \mathfrak{R}_\phi^\pi)$, is equal to the expected return $J(\pi; M)$ of $\pi$ by Theorem 3.1, which is bounded.

# B Proofs of Theoretical Results

In this section, we provide proofs of the theorems and lemmas in the main text. We start with Theorem 3.1 that states that ARPs are *performance-preserving*.

## B.1 ARPs are Performance Preserving

**Theorem 3.1.** $\forall \phi \in \Phi$, *the performance of a policy $\pi$ is equal to the expected return of the abstract reward process $\mathfrak{R}_\phi^\pi$ defined from MDP $M$, i.e., $J(\pi; \mathfrak{R}_\phi^\pi) = J(\pi; M)$.*

*Proof.* This result states that despite the use of a discrete state abstraction, the expected return of the ARP $\mathfrak{R}_\phi^\pi$ is equal to the performance of policy $\pi$. To prove this, we leverage two equivalent forms of defining the expected return of a policy. The first form:

$$J(\pi; \mathfrak{R}_\phi^\pi) = \sum_t \mathbb{E}[R_t; \pi], \tag{7}$$

defines the expected return as a sum of the expected value of the reward at each timestep, where the distribution of rewards at each timestep is governed by the components of the ARP, namely, $\eta_\phi, \mathrm{P}_\phi^\pi$ and $\mathrm{R}_\phi^\pi$. The second, equivalent, form of expressing the expected return is given by:

$$J(\pi; \mathfrak{R}_\phi^\pi) = \sum_t \sum_{z \in \mathcal{Z}} \Pr(Z_t = z; \pi)\mathbb{E}[R_t | Z_t {=} z; \pi] = \sum_t \sum_{z \in \mathcal{Z}} \Pr(Z_t = z; \pi)\mathrm{R}_\phi^\pi(z), \tag{8}$$

where the expected return is denoted as a sum of the reward at each abstract state mutliplied by the visitation frequency of that abstract state. Both Equations 7 and 8 are equivalent, i.e.,

$$J(\pi; \mathfrak{R}_\phi^\pi) = \sum_t \mathbb{E}[R_t; \pi] = \sum_t \sum_{z \in \mathcal{Z}} \Pr(Z_t = z; \pi)\mathrm{R}_\phi^\pi(z). \tag{9}$$

Next, we denote the terms in Equation 8 in terms of the states of the underlying MDP to show the final result. The reward function $\mathrm{R}_\phi^\pi$ can be expressed as,

$$\begin{aligned}
\mathrm{R}_\phi^\pi(z) &:= \frac{\sum_t \mathbb{E}[R_t | \phi(S_t){=}z; \pi] \Pr(\phi(S_t){=}z; \pi)}{\sum_t \Pr(\phi(S_t){=}z; \pi)} \\
&= \frac{\sum_t \sum_{s \in \mathcal{S}} \mathbf{1}\{\phi(s) = z\}\mathbb{E}[R_t | S_t{=}s; \pi] \Pr(S_t{=}s; \pi)}{\sum_t \sum_{s \in \mathcal{S}} \mathbf{1}\{\phi(s) = z\} \Pr(S_t{=}s; \pi)} \\
&= \frac{\sum_t \sum_{s \in \mathcal{S}, a \in \mathcal{A}} \mathbf{1}\{\phi(s) = z\}r(s, a)\pi(s, a) \Pr(S_t{=}s; \pi)}{\sum_t \sum_{s \in \mathcal{S}} \mathbf{1}\{\phi(s) = z\} \Pr(S_t{=}s; \pi)} \\
&= \frac{\sum_{s \in \mathcal{S}, a \in \mathcal{A}} \mathbf{1}\{\phi(s) = z\}r(s, a)\pi(s, a)(\sum_t \Pr(S_t{=}s; \pi))}{\sum_{s \in \mathcal{S}} \mathbf{1}\{\phi(s) = z\}(\sum_t \Pr(S_t{=}s; \pi))} \\
&= \frac{\sum_{s \in \mathcal{S}, a \in \mathcal{A}} \psi^\pi(s)\pi(s, a)r(s, a)\mathbf{1}\{\phi(s) = z\}}{\sum_{s \in \mathcal{S}} \psi^\pi(s)\mathbf{1}\{\phi(s) = z\}},
\end{aligned}$$

where the undiscounted state distribution [49, Section 9.2] under policy $\pi$ is denoted by $\psi^\pi(s) \propto \sum_t \Pr(S_t = s; \pi)$, where $\psi^\pi(s_\infty)$ is set to be proportional to any constant value. The normalization constant, $\kappa$, that makes $\psi^\pi(s)$ a valid distribution cancels out from both the numerator and the denominator of the above expression. The term $\Pr(Z_t = z; \pi)$ can be expressed in terms of states as,

$$\Pr(Z_t = z; \pi) = \sum_{s \in \mathcal{S}} \Pr(S_t = s; \pi)\mathbf{1}\{\phi(s) = z\}.$$

Substituting these expressions into Equation 8 gives,

$$
\begin{aligned}
J(\pi; \mathfrak{R}_\phi^\pi) &= \sum_t \sum_{z \in \mathcal{Z}} \Pr(Z_t = z; \pi) \mathrm{R}_\phi^\pi(z) \\
&= \sum_t \sum_{z \in \mathcal{Z}} \left( \sum_{s \in \mathcal{S}} \Pr(S_t = s; \pi) \mathbf{1}\{\phi(s) = z\} \right) \left( \frac{\sum_{s \in \mathcal{S}, a \in \mathcal{A}} \psi^\pi(s) \pi(s,a) r(s,a) \mathbf{1}\{\phi(s) = z\}}{\sum_{s \in \mathcal{S}} \psi^\pi(s) \mathbf{1}\{\phi(s) = z\}} \right) \\
&= \sum_{z \in \mathcal{Z}} \left( \sum_{s \in \mathcal{S}} \sum_t \Pr(S_t = s; \pi) \mathbf{1}\{\phi(s) = z\} \right) \left( \frac{\sum_{s \in \mathcal{S}, a \in \mathcal{A}} \psi^\pi(s) \pi(s,a) r(s,a) \mathbf{1}\{\phi(s) = z\}}{\sum_{s \in \mathcal{S}} \psi^\pi(s) \mathbf{1}\{\phi(s) = z\}} \right) \\
&= \sum_{z \in \mathcal{Z}} \left( \sum_{s \in \mathcal{S}} \kappa \, \psi^\pi(s) \mathbf{1}\{\phi(s) = z\} \right) \left( \frac{\sum_{s \in \mathcal{S}, a \in \mathcal{A}} \psi^\pi(s) \pi(s,a) r(s,a) \mathbf{1}\{\phi(s) = z\}}{\sum_{s \in \mathcal{S}} \psi^\pi(s) \mathbf{1}\{\phi(s) = z\}} \right) \\
&= \sum_{z \in \mathcal{Z}} \sum_{s \in \mathcal{S}, a \in \mathcal{A}} \kappa \, \psi^\pi(s) \pi(s,a) r(s,a) \mathbf{1}\{\phi(s) = z\} \\
&\overset{(a)}{=} \sum_{s \in \mathcal{S}, a \in \mathcal{A}} \sum_t \Pr(S_t = s; \pi) \pi(s,a) r(s,a) \\
&= J(\pi; M),
\end{aligned}
$$

where (a) follows from the law of total probability [64].

$\square$

We have established that ARPs are performance-preserving. Next, we prove Lemma 3.2, which states that the performance estimate obtained from an estimated model of the ARP converges almost surely to the performance of the policy that induces it. In order to do so, we first introduce properties that will be useful in the proof of Lemma 3.2.

## B.2 Estimation of ARPs from On-Policy Data

At a high level, we prove Lemma 3.2 by first studying the almost sure convergence of the estimated transition function, reward function, and initial state distributions. Once the almost sure convergence of these components has been established, we reason about the implications for the convergence of the policy performance predictions that result from these terms. We begin by studying the almost sure convergence of the transition function.

**Property B.2.** *For all abstract states $z \in \mathcal{Z}$ and $z' \in \mathcal{Z}$, if*

$$
\sum_t \Pr(Z_t = z; \pi) \neq 0,
$$

*then*

$$
\widehat{\mathrm{P}}_{n,\phi}^\pi(z,z') \overset{a.s.}{\longrightarrow} \mathrm{P}_\phi^\pi(z,z'). \tag{10}
$$

*Proof.* Recall that

$$
\widehat{\mathrm{P}}_{n,\phi}^\pi(z,z') = \frac{\sum_{i=1}^n \sum_t \mathbf{1}\{Z_t^i = z, Z_{t+1}^i = z'\}}{\sum_{i=1}^n \sum_t \mathbf{1}\{Z_t^i = z\}}.
$$

Let $X_n := \frac{1}{n} \sum_{i=1}^n \sum_t \mathbf{1}\{Z_t^i = z, Z_{t+1}^i = z'\}$ and $Y_n := \frac{1}{n} \sum_{i=1}^n \sum_t \mathbf{1}\{Z_t^i = z\}$. We can then rewrite $\widehat{\mathrm{P}}_{n,\phi}^\pi(z,z')$ as

$$
\widehat{\mathrm{P}}_{n,\phi}^\pi(z,z') = \frac{X_n}{Y_n},
$$

which is a continuous function of $X_n$ and $Y_n$ when $Y_n > 0$. At a high level, we will show what $X_n$ and $Y_n$ each converge to almost surely, and will then apply the continuous mapping theorem to reason about the almost sure convergence of $\widehat{\mathrm{P}}_{n,\phi}^\pi(z,z')$.

However, there may be some abstract states $\tilde{Z} \subset \mathcal{Z}$ that are not reached and thus are not observed in the data. Such abstract states pose a problem: If $\tilde{z} \in \tilde{Z}$, then $Y_n$ for $\widehat{\mathrm{P}}_{n,\phi}^\pi(\tilde{z},z')$ will be zero,

and so $\widehat{\mathrm{P}}_{n,\phi}^{\pi}(\tilde{z}, z')$ will be undefined and not a continuous function of $X_n$ and $Y_n$ (which must be handled appropriately in the proof of consistency). First, to ensure that the ARP is well-defined even when $\tilde{Z}$ is not empty, for abstract states $\tilde{z} \in \tilde{Z}$ (abstract states that were not observed in the data), special values can be hardcoded into the transition function, reward functions, and initial state distribution so that the components of the ARP continue to be well-defined.[4] In particular, for $\tilde{z} \in \tilde{Z}$, the transition function can be set to self-transition back to $\tilde{z}$ with probability 1, i.e., $\widehat{\mathrm{P}}_{n,\phi}^{\pi}(\tilde{z}, \tilde{z}) = 1$ and $\widehat{\mathrm{P}}_{n,\phi}^{\pi}(\tilde{z}, \bar{z}) = 0$ for all $\bar{z} \notin \tilde{Z}$, the reward function is set to $\widehat{\mathrm{R}}_{n,\phi}^{\pi}(\tilde{z}) = 0$ and the initial abstract state distribution is $\widehat{\eta}_{n,\phi}^{\pi}(\tilde{z}) = 0$ for all $\tilde{z} \in \tilde{Z}$.[5]

To reason about the almost sure convergence of $\widehat{\mathrm{P}}_{n,\phi}^{\pi}(z, z')$, first consider $\lim_{n \to \infty} X_n$,

$$\lim_{n \to \infty} X_n = \lim_{n \to \infty} \frac{1}{n} \sum_{i=1}^{n} \sum_{t} \mathbf{1}\{Z_t^i = z, Z_{t+1}^i = z'\}$$

$$= \lim_{n \to \infty} \sum_{t} \left( \frac{1}{n} \sum_{i=1}^{n} \mathbf{1}\{Z_t^i = z, Z_{t+1}^i = z'\} \right).$$

Note that $\mathbf{1}\{Z_t^i = z, Z_{t+1}^i = z'\}$ is independent across episodes (for each $i$), has bounded variance, and has the same mean for each episode. By Kolmogorov's strong law of large numbers [46],

$$\frac{1}{n} \sum_{i=1}^{n} \mathbf{1}\{Z_t^i = z, Z_{t+1}^i = z'\} \xrightarrow{\text{a.s.}} \mathbb{E}\left[\mathbf{1}\{Z_t = z, Z_{t+1} = z'\}; \pi\right],$$

This convergence guarantee holds for all $t$. From the definition of almost sure convergence, this means that

$$\forall t \in \mathbb{N}, \Pr\left(\lim_{n \to \infty} \frac{1}{n} \sum_{i=1}^{n} \mathbf{1}\{Z_t^i = z, Z_{t+1}^i = z'\} = \mathbb{E}\left[\mathbf{1}\{Z_t = z, Z_{t+1} = z'\}; \pi\right]\right) = 1, \quad (11)$$

where the expectation results from the use of the policy $\pi$ that generated dataset $\mathcal{D}_n^{(\pi)}$. By the countable additivity property of probability measures (and the fact that the number of time steps is countable), this implies the following (notice that $\forall t \in \mathbb{N}$ is now inside the statement of probability):

$$\Pr\left(\forall t \in \mathbb{N}, \lim_{n \to \infty} \frac{1}{n} \sum_{i=1}^{n} \mathbf{1}\{Z_t^i = z, Z_{t+1}^i = z'\} = \mathbb{E}\left[\mathbf{1}\{Z_t = z, Z_{t+1} = z'\}; \pi\right]\right) = 1. \quad (12)$$

Hence,

$$\Pr\left(\sum_{t} \lim_{n \to \infty} \frac{1}{n} \sum_{i=1}^{n} \mathbf{1}\{Z_t^i = z, Z_{t+1}^i = z'\} = \sum_{t} \mathbb{E}\left[\mathbf{1}\{Z_t = z, Z_{t+1} = z'\}; \pi\right]\right) = 1. \quad (13)$$

Next, by the dominated convergence theorem [16, Theorem 1.19], the summation over time and limit over $n$ can be interchanged, giving:

$$\Pr\left(\lim_{n \to \infty} \sum_{t} \frac{1}{n} \sum_{i=1}^{n} \mathbf{1}\{Z_t^i = z, Z_{t+1}^i = z'\} = \sum_{t} \mathbb{E}\left[\mathbf{1}\{Z_t = z, Z_{t+1} = z'\}; \pi\right]\right) = 1. \quad (14)$$

Returning to $\xrightarrow{\text{a.s.}}$ notation, this means that

$$\sum_{t} \left( \frac{1}{n} \sum_{i=1}^{n} \mathbf{1}\{Z_t^i = z, Z_{t+1}^i = z'\} \right) \xrightarrow{\text{a.s.}} \sum_{t} \mathbb{E}\left[\mathbf{1}\{Z_t = z, Z_{t+1} = z'\}; \pi\right]. \quad (15)$$

---

[4]The particular behavior of the ARP in these unobserved abstract states is not of consequence. Since they are unreached, the behavior in these abstract states does not impact the expected return of the ARP. However, we wish to ensure that the transition function, reward function, and initial state distribution still represent a well-defined ARP.

[5]Note that the initial state distribution is not actually a problem like the transition and reward functions—the previous definitions already ensured that $\widehat{\eta}_{n,\phi}^{\pi}(\tilde{z}) = 0$.

The expected value of the indicator is equal to the probability of the event, and so

$$\sum_t \mathbb{E}\left[\mathbf{1}\{Z_t = z, Z_{t+1} = z'\}; \pi\right] = \sum_t \Pr(Z_t = z, Z_{t+1} = z'; \pi). \tag{16}$$

This, combined with the fact that the left hand side of Equation 15 is $X_n$ means that

$$X_n \xrightarrow{\text{a.s.}} \sum_t \Pr(Z_t = z, Z_{t+1} = z'; \pi). \tag{17}$$

Similarly, these same exact steps can be followed to show that

$$Y_n \xrightarrow{\text{a.s.}} \sum_t \Pr(Z_t = z; \pi). \tag{18}$$

Let $\Psi_i = (X_i, Y_i)$, so that $\Psi_n = (X_i, Y_i)_{i=1}^n$ is a sequence of vector-valued random variables. Since $X_n \xrightarrow{\text{a.s.}} \sum_t \Pr(Z_t = z, Z_{t+1} = z'; \pi)$ and $Y_n \xrightarrow{\text{a.s.}} \sum_t \Pr(Z_t = z; \pi)$, we have that

$$\Psi_n \xrightarrow{\text{a.s.}} \left(\sum_t \Pr(Z_t = z, Z_{t+1} = z'; \pi), \sum_t \Pr(Z_t = z; \pi)\right). \tag{19}$$

Consider the function $f : \mathbb{R}^2 \to \mathbb{R}$, defined as:

$$f(x, y) = \begin{cases} \frac{x}{y} & \text{if } y \neq 0 \\ \widehat{\mathrm{P}}_{n,\phi}^\pi(z, z') & \text{otherwise.} \end{cases} \tag{20}$$

Note that discontinuities of $f$ may occur when $y = 0$. Applying the continuous mapping theorem, we have that

$$f(\Psi_n) \xrightarrow{\text{a.s.}} \frac{\sum_t \Pr(Z_t = z, Z_{t+1} = z'; \pi)}{\sum_t \Pr(Z_t = z; \pi)} \tag{21}$$

if[6]

$$\Pr\left(\sum_t \Pr(Z_t = z; \pi) = 0\right) = 0. \tag{22}$$

In our case, $\sum_t \Pr(Z_t = z; \pi)$ is a constant, and so the condition in Equation 22 is simply:

$$\sum_t \Pr(Z_t = z; \pi) \neq 0. \tag{23}$$

Also, Equation 21 can be simplified to:

$$\widehat{\mathrm{P}}_{n,\phi}^\pi(z, z') \xrightarrow{\text{a.s.}} \mathrm{P}_\phi^\pi(z, z'), \tag{24}$$

since $f(\Psi_n) = X_n/Y_n = \widehat{\mathrm{P}}_{n,\phi}^\pi(z, z')$,[7] and $\mathrm{P}_\phi^\pi(z, z') = \frac{\sum_t \Pr(Z_t = z, Z_{t+1} = z'; \pi)}{\sum_t \Pr(Z_t = z; \pi)}$ by Equation 3. Restating this conclusion, we have that if

$$\sum_t \Pr(Z_t = z; \pi) \neq 0, \tag{25}$$

then

$$\widehat{\mathrm{P}}_{n,\phi}^\pi(z, z') \xrightarrow{\text{a.s.}} \mathrm{P}_\phi^\pi(z, z'), \tag{26}$$

which establishes the property. $\qquad\square$

Next, we present a property that establishes the almost sure convergence of the reward function estimate.

---

[6]This condition arises due to the discontinuity of $f$ when the second argument is zero. See the statement of the continuous mapping theorem in Appendix A for details.

[7]To be more precise $f(\Psi_n) = X_n/Y_n$ when $Y_n \neq 0$. However, even when $Y_n = 0$, the conclusion that $f(\Psi_n) = \widehat{\mathrm{P}}_{n,\phi}^\pi(z, z')$ holds from the definition of $f$ in Equation 20.

**Property B.3.** *For all abstract states* $z \in \mathcal{Z}$, *if*

$$\sum_t \Pr(Z_t = z; \pi) \neq 0, \tag{27}$$

*then*

$$\widehat{R}^\pi_{n,\phi}(z) \xrightarrow{a.s.} R^\pi_\phi(z). \tag{28}$$

*Proof.* Recall that

$$\widehat{R}^\pi_{n,\phi}(z) = \frac{\sum_{i=1}^n \sum_t \mathbf{1}\{Z^i_t = z\} R^i_t}{\sum_{i=1}^n \sum_t \mathbf{1}\{Z^i_t = z\}}.$$

Let $X_n := \frac{1}{n} \sum_{i=1}^n \sum_t \mathbf{1}\{Z^i_t = z\} R^i_t$ and $Y_n := \frac{1}{n} \sum_{i=1}^n \sum_t \mathbf{1}\{Z^i_t = z\}$. We will show that $X_n$ and $Y_n$ converge almost surely, and then apply the continuous mapping theorem to reason about the almost sure convergence of $\widehat{R}^\pi_{n,\phi}(z)$.

To ensure that the ARP is well-defined even when $\tilde{Z}$ is not empty, for abstract states $\tilde{z} \in \tilde{Z}$ (abstract states that were not observed in the data), the reward function can be set to $\widehat{R}^\pi_{n,\phi}(\tilde{z}) = 0$, as elaborated in the proof of Property B.2.

Consider $\lim_{n\to\infty} X_n$,

$$\lim_{n\to\infty} X_n = \lim_{n\to\infty} \frac{1}{n} \sum_{i=1}^n \sum_t \mathbf{1}\{Z^i_t = z\} R^i_t$$

$$= \lim_{n\to\infty} \sum_t \left( \frac{1}{n} \sum_{i=1}^n \mathbf{1}\{Z^i_t = z\} R^i_t \right).$$

Note that $\mathbf{1}\{Z^i_t = z\} R^i_t$ is independent across episodes (for each $i$), has bounded variance, and has the same mean for each episode. By Kolmogorov's strong law of large numbers [46],

$$\frac{1}{n} \sum_{i=1}^n \mathbf{1}\{Z^i_t = z\} R^i_t \xrightarrow{a.s.} \mathbb{E}\left[ \mathbf{1}\{Z_t = z\} R_t; \pi \right]$$

$$= \sum_{r \in \mathbb{R}, \check{z} \in \mathcal{Z}} \Pr(R_t = r, Z_t = \check{z}; \pi) \mathbf{1}\{Z_t = z\} r$$

$$= \sum_{r \in \mathbb{R}} \Pr(R_t = r, Z_t = z; \pi) r$$

$$= \left( \sum_{r \in \mathbb{R}} r \Pr(R_t = r \mid Z_t = z; \pi) \right) \Pr(Z_t = z; \pi)$$

$$= \mathbb{E}\left[ R_t \mid Z_t = z; \pi \right] \Pr(Z_t = z; \pi).$$

This convergence guarantee holds for all $t$. As in the proof of Property B.2, by the countable additivity property of probability measures, in conjunction with the dominated convergence theorem [16, Theorem 1.19], this implies that,

$$\Pr\left( \lim_{n\to\infty} \sum_t \frac{1}{n} \sum_{i=1}^n \mathbf{1}\{Z^i_t = z\} R^i_t = \sum_t \mathbb{E}\left[ R_t \mid Z_t = z; \pi \right] \Pr(Z_t = z; \pi) \right) = 1. \tag{29}$$

Returning to $\xrightarrow{a.s.}$ notation, this means that

$$X_n \xrightarrow{a.s.} \sum_t \mathbb{E}\left[ R_t \mid Z_t = z; \pi \right] \Pr(Z_t = z; \pi). \tag{30}$$

Similarly, these same exact steps can be followed to show that

$$Y_n \xrightarrow{a.s.} \sum_t \Pr(Z_t = z; \pi). \tag{31}$$

Let $\Psi_i = (X_i, Y_i)$, so that $\Psi_n = (X_i, Y_i)_{i=1}^n$ is a sequence of vector-valued random variables. Considering the function $f : \mathbb{R}^2 \to \mathbb{R}$, defined as:

$$f(x, y) = \begin{cases} \frac{x}{y} & \text{if } y \neq 0 \\ \widehat{\mathrm{R}}_{n,\phi}^{\pi}(z) & \text{otherwise.} \end{cases} \tag{32}$$

Note that discontinuities of $f$ may occur when $y = 0$. Applying the continuous mapping theorem, we have that

$$f(\Psi_n) \xrightarrow{\text{a.s.}} \frac{\sum_t \mathbb{E}\left[R_t \mid Z_t = z; \pi\right] \Pr(Z_t = z; \pi)}{\sum_t \Pr(Z_t = z; \pi)} \tag{33}$$

if

$$\sum_t \Pr(Z_t = z; \pi) \neq 0. \tag{34}$$

Thus, when $\sum_t \Pr(Z_t = z; \pi) \neq 0$

$$\widehat{\mathrm{R}}_{n,\phi}^{\pi}(z) \xrightarrow{\text{a.s.}} \mathrm{R}_{\phi}^{\pi}(z), \tag{35}$$

which establishes the property. $\qquad\square$

Next, we present a property that establishes the almost sure convergence of the initial state distribution.

**Property B.4.** *For all abstract states $z \in \mathcal{Z}$, $\widehat{\eta}_{n,\phi}(z) \xrightarrow{\text{a.s.}} \eta_\phi(z)$.*

*Proof.* Recall that

$$\widehat{\eta}_{n,\phi}(z) = \frac{1}{n} \sum_{i=1}^n \mathbf{1}\{Z_0^i = z\}.$$

Let $X_n := \frac{1}{n} \sum_{i=1}^n \mathbf{1}\{Z_0^i = z\}$. We will show that $X_n$ converges almost surely, and then apply the continuous mapping theorem to reason about the almost sure convergence of $\widehat{\eta}_{n,\phi}(z)$. Note that even when $\tilde{Z}$ is not empty, the definition of $\widehat{\eta}_{n,\phi}^{\pi}$ ensures that it is well-defined.

Consider $\lim_{n\to\infty} X_n$,

$$\lim_{n\to\infty} X_n = \lim_{n\to\infty} \frac{1}{n} \sum_{i=1}^n \mathbf{1}\{Z_0^i = z\}.$$

Note that $\mathbf{1}\{Z_0^i = z\}$ is independent across episodes (for each $i$), has bounded variance, and has the same mean for each episode. By Kolmogorov's strong law of large numbers [46],

$$\frac{1}{n} \mathbf{1}\{Z_0^i = z\} \xrightarrow{\text{a.s.}} \mathbb{E}\left[\mathbf{1}\{Z_0 = z\}\right] = \Pr(Z_0 = z).$$

In contrast to the proofs of the previous two properties, this proof holds for all $z \in \mathcal{Z}$, not just the abstract states that have non-zero probability of occuring. This implies that

$$\widehat{\eta}_{n,\phi}(z) \xrightarrow{\text{a.s.}} \eta_\phi(z), \tag{36}$$

which establishes the property. $\qquad\square$

Having established properties that will be useful in the proof of Lemma 3.2, we now turn to proving the lemma.

**Lemma 3.2.** $\forall \phi \in \Phi$, *the expected return of the maximum likelihood estimate $\widehat{\mathfrak{R}}_{n,\phi}^{\pi}$ converges almost surely to the expected return of the policy $\pi$, i.e., $J(\pi; \widehat{\mathfrak{R}}_{n,\phi}^{\pi}) \xrightarrow{\text{a.s.}} J(\pi; M)$.*

*Proof.* From Theorem 3.1, we have that $J(\pi; \mathfrak{R}_{\phi}^{\pi}) = J(\pi; M)$. Thus, to prove this result, we only need to show that $J(\pi; \widehat{\mathfrak{R}}_{n,\phi}^{\pi}) \xrightarrow{\text{a.s.}} J(\pi; \mathfrak{R}_{\phi}^{\pi})$. Several of the preliminary results required to estalish this result were provided in Properties B.2, B.3, and B.4. Specifically, Properties B.2 and B.3 establish that for all abstract states $z \in \mathcal{Z}$ and $z' \in \mathcal{Z}$, if

$$\sum_t \Pr(Z_t = z; \pi) \neq 0, \tag{37}$$

then

$$\widehat{P}^{\pi}_{n,\phi}(z, z') \xrightarrow{\text{a.s.}} P^{\pi}_{\phi}(z, z'), \tag{38}$$

and

$$\widehat{R}^{\pi}_{n,\phi}(z) \xrightarrow{\text{a.s.}} R^{\pi}_{\phi}(z). \tag{39}$$

Similarly, Property B.4 establishes that for all abstract states $z \in \mathcal{Z}$,

$$\widehat{\eta}_{n,\phi}(z) \xrightarrow{\text{a.s.}} \eta_{\phi}(z). \tag{40}$$

The expected return $J(\pi; \widehat{\mathfrak{R}}^{\pi}_{n,\phi})$ is a continuous function of $\widehat{\mathfrak{R}}^{\pi}_{n,\phi} := (\widehat{P}^{\pi}_{n,\phi}, \widehat{R}^{\pi}_{n,\phi}, \widehat{\eta}_{n,\phi})$, given by $J(\pi; \widehat{\mathfrak{R}}^{\pi}_{n,\phi}) = \left(I - \widehat{P}^{\pi}_{n,\phi}\right)^{-1} \widehat{R}^{\pi}_{n,\phi} \widehat{\eta}_{n,\phi} = \sum_z \sum_t \hat{\Pr}(Z_t = z; \pi) \widehat{R}^{\pi}_{n,\phi}(z)$[8] where $\hat{\Pr}$ denotes the empirical estimate of the corresponding probability.

Note that the term $\sum_t \hat{\Pr}(Z_t = z; \pi) = \left[ \left(I - \widehat{P}^{\pi}_{n,\phi}\right)^{-1} \widehat{\eta}_{n,\phi} \right]_z$, i.e., the empirical estimate of the sum of probabilities of encountering the abstract state $z$ over all timesteps is equal to value of the vector $\left(I - \widehat{P}^{\pi}_{n,\phi}\right)^{-1} \widehat{\eta}_{n,\phi}$ at $z$. By the continuous mapping theorem, if $\sum_t \Pr(Z_t = z; \pi) \neq 0$,

$$\sum_t \hat{\Pr}(Z_t = z; \pi) \xrightarrow{\text{a.s.}} \sum_t \Pr(Z_t = z; \pi). \tag{41}$$

Let $\bar{Z} = \{z : \sum_t \Pr(Z_t = z; \pi) = 0\}$, where $\bar{Z} \subset \mathcal{Z}$, be the set of abstract states that will never be observed empirically as they have no probability of being visited under $\pi$.[9] The expression for the expected return can be divided into two terms: (1) the first term sums over abstract states for which $\sum_t \Pr(Z_t = z; \pi) = 0$, and the values of the components of the ARP—in particular, $\widehat{R}^{\pi}_{n,\phi}$—have been specially hardcoded as detailed in Property B.2, and (2) the second term sums over the abstract states for which $\sum_t \Pr(Z_t = z; \pi) \neq 0$ and thus Equation 41 holds.

$$J(\pi; \widehat{\mathfrak{R}}^{\pi}_{n,\phi}) = \sum_{z \in \bar{Z}} \sum_t \hat{\Pr}(Z_t = z; \pi) \underbrace{\widehat{R}^{\pi}_{n,\phi}(z)}_{=0} + \sum_{z \in \mathcal{Z} \setminus \bar{Z}} \sum_t \hat{\Pr}(Z_t = z; \pi) \widehat{R}^{\pi}_{n,\phi}(z) \tag{42}$$

$$= \sum_{z \in \mathcal{Z} \setminus \bar{Z}} \sum_t \hat{\Pr}(Z_t = z; \pi) \widehat{R}^{\pi}_{n,\phi}(z) \tag{43}$$

As detailed in Property B.2, $\widehat{R}^{\pi}_{n,\phi}(z) = 0$ for $z \in \tilde{Z}$ and thus $\forall z \in \bar{Z}$, making the first term equal to zero. The constituents of the second term converge almost surely:

$$\sum_{z \in \mathcal{Z} \setminus \bar{Z}} \sum_t \hat{\Pr}(Z_t = z; \pi) \widehat{R}^{\pi}_{n,\phi}(z) \xrightarrow{\text{a.s.}} \sum_{z \in \mathcal{Z} \setminus \bar{Z}} \sum_t \Pr(Z_t = z; \pi) R^{\pi}_{\phi}(z) \tag{44}$$

$$= \sum_{z \in \mathcal{Z} \setminus \bar{Z}} \sum_t \Pr(Z_t = z; \pi) R^{\pi}_{\phi}(z) + 0 \tag{45}$$

$$= \sum_{z \in \mathcal{Z} \setminus \bar{Z}} \sum_t \Pr(Z_t = z; \pi) R^{\pi}_{\phi}(z) + \sum_{z \in \bar{Z}} \underbrace{\left( \sum_t \Pr(Z_t = z; \pi) \right)}_{=0} R^{\pi}_{\phi}(z) \tag{46}$$

$$= \sum_{z \in \mathcal{Z}} \sum_t \Pr(Z_t = z; \pi) R^{\pi}_{\phi}(z) \tag{47}$$

$$= J(\pi; \mathfrak{R}^{\pi}_{\phi}). \tag{48}$$

We then obtain the final result:

$$J(\pi; \widehat{\mathfrak{R}}^{\pi}_{n,\phi}) \xrightarrow{\text{a.s.}} J(\pi; \mathfrak{R}^{\pi}_{\phi}) = J(\pi; M). \tag{49}$$

$\square$

---

[8]Termination is handled as detailed in Appendix A.1.

[9]In relation to $\tilde{Z}$ defined in Property B.2, note $\bar{Z} \subseteq \tilde{Z}$. As $n \to \infty$ the two sets, with probability 1, become equal.

## B.3  Estimation of ARPs from Off-Policy Data

So far, we have shown that the expected return of the estimated ARP converges almost surely to the expected return of the policy that induced it, i.e., consistent *on-policy* evaluation. Next, we show that a model of the ARP can be consistently estimated from *off-policy* data. This requires us to prove that transition functions estimated using off-policy data, incorporating importance weights in their estimation, converge almost surely to the true transition functions induced by the evaluation policy, and that the same holds for the reward functions and initial state distributions.

**Property B.6.** *For all abstract states $z \in \mathcal{Z}$ and $z' \in \mathcal{Z}$, if*

$$\sum_t \Pr(Z_t = z; \pi_b) \neq 0, \tag{50}$$

*then*

$$\widehat{P}_{n,\phi}^{\pi_b \to \pi_e}(z, z') \xrightarrow{a.s.} P_{\phi}^{\pi_e}(z, z'). \tag{51}$$

*Proof.* Recall that

$$\widehat{P}_{n,\phi}^{\pi_b \to \pi_e}(z, z') = \frac{\sum_{i,t} \mathbf{1}_t^i \{z, z'\} \rho_{0:t}}{\sum_{i,t} \mathbf{1}_t^i \{z\} \rho_{0:t}}.$$

Similar to the proof structure of Property B.2, let $X_n := \frac{1}{n} \sum_{i,t} \mathbf{1}_t^i \{z, z'\} \rho_{0:t}$ and $Y_n := \frac{1}{n} \sum_{i,t} \mathbf{1}_t^i \{z\} \rho_{0:t}$. For abstract states $\tilde{Z} \subset \mathcal{Z}$ that are never reached and thus are not observed in the data, special values can be hardcoded into the transition function, reward functions, and initial state distribution so that the components of the ARP continue to be well-defined. In particular, for $\tilde{z} \in \tilde{Z}$, the transition function can be set to self-transition back to $\tilde{z}$ with probability 1, i.e., $\widehat{P}_{n,\phi}^{\pi_b \to \pi_e}(\tilde{z}, \tilde{z}) = 1$ and $\widehat{P}_{n,\phi}^{\pi_b \to \pi_e}(\tilde{z}, \bar{z}) = 0$ for all $\bar{z} \notin \tilde{Z}$, the reward function is set to $\widehat{R}_{n,\phi}^{\pi_b \to \pi_e}(\tilde{z}) = 0$ and the initial abstract state distribution is $\widehat{\eta}_{n,\phi}^{\pi_b \to \pi_e}(\tilde{z}) = 0$ for all $\tilde{z} \in \tilde{Z}$.

Consider $\lim_{n \to \infty} X_n$,

$$\lim_{n \to \infty} X_n = \lim_{n \to \infty} \frac{1}{n} \sum_{i,t} \mathbf{1}_t^i \{z, z'\} \rho_{0:t}$$

$$= \lim_{n \to \infty} \sum_t \left( \frac{1}{n} \sum_{i=1}^n \mathbf{1}_t^i \{z, z'\} \rho_{0:t} \right).$$

Note that $\mathbf{1}_t^i \{z, z'\} \rho_{0:t}$ is independent across episodes (for each $i$), has bounded variance, and has the same mean for each episode. By Kolmogorov's strong law of large numbers [46],

$$\frac{1}{n} \sum_{i=1}^n \mathbf{1}_t^i \{z, z'\} \rho_{0:t} \xrightarrow{a.s.} \mathbb{E}\left[ \mathbf{1}\{Z_t = z, Z_{t+1} = z'\} \rho_{0:t}; \pi_b \right].$$

The importance weights change the distribution over which the expectation is computed, as below:

$$\mathbb{E}\left[ \mathbf{1}\{Z_t = z, Z_{t+1} = z'\} \rho_{0:t}; \pi_b \right]$$

$$= \sum_t \sum_{\substack{S_{0:t+1}, \\ A_{0:t}}} \left( \eta(S_0) \prod_{l=0}^t \pi_b(S_l, A_l) p(S_l, A_l, S_{l+1}) \right) \mathbf{1}\{\phi(S_t)=z, \phi(S_{t+1})=z'\} \left( \prod_{j=0}^t \frac{\pi_e(S_j, A_j)}{\pi_b(S_j, A_j)} \right)$$

$$= \sum_t \sum_{\substack{S_{0:t+1}, \\ A_{0:t}}} \left( \eta(S_0) \prod_{l=0}^t \pi_e(S_l, A_l) p(S_l, A_l, S_{l+1}) \right) \mathbf{1}\{\phi(S_t)=z, \phi(S_{t+1})=z'\}$$

$$= \sum_t \mathbb{E}\left[ \mathbf{1}\{Z_t = z, Z_{t+1} = z'\}; \pi_e \right]$$

This convergence guarantee holds for all $t$. By the countable additivity property of probability measures in conjunction with the dominated convergence theorem [16], this implies that,

$$X_n \xrightarrow{a.s.} \sum_t \mathbb{E}\left[ \mathbf{1}\{Z_t = z, Z_{t+1} = z'\}; \pi_e \right]. \tag{52}$$

Similarly, these same exact steps can be followed to show that

$$Y_n \xrightarrow{\text{a.s.}} \sum_t \mathbb{E}\left[\mathbf{1}\{Z_t = z\}; \pi_e\right]. \tag{53}$$

Let $\Psi_i = (X_i, Y_i)$, so that $\Psi_n = (X_i, Y_i)_{i=1}^n$ is a sequence of vector-valued random variables. Considering the function $f : \mathbb{R}^2 \to \mathbb{R}$, defined as:

$$f(x, y) = \begin{cases} \frac{x}{y} & \text{if } y \neq 0 \\ \widehat{\text{P}}_{n,\phi}^{\pi_b \to \pi_e}(z, z') & \text{otherwise.} \end{cases} \tag{54}$$

Note that discontinuities of $f$ may occur when $y = 0$. Applying the continuous mapping theorem, we have that

$$f(\Psi_n) \xrightarrow{\text{a.s.}} \frac{\sum_t \mathbb{E}\left[\mathbf{1}\{Z_t = z, Z_{t+1} = z'\}; \pi_e\right]}{\sum_t \mathbb{E}\left[\mathbf{1}\{Z_t = z\}; \pi_e\right]} \tag{55}$$

if $\sum_t \mathbb{E}\left[\mathbf{1}\{Z_t = z\}; \pi_e\right] \neq 0$. This implies that

$$\widehat{\text{P}}_{n,\phi}^{\pi_b \to \pi_e}(z, z') \xrightarrow{\text{a.s.}} \text{P}_\phi^{\pi_e}(z, z'), \tag{56}$$

when $\sum_t \Pr(Z_t = z; \pi_b) \neq 0$, which establishes the property.

$\square$

Next, we present the properties that establish almost sure convergence of the reward function estimate and the initial state distribution estimated from off-policy data. This follows a similar proof structure to their corresponding on-policy properties, with the key difference being the use of importance weights in the estimation.

**Property B.7.** *For all abstract states $z \in \mathcal{Z}$, if*

$$\sum_t \Pr(Z_t = z; \pi_b) \neq 0, \tag{57}$$

*then*

$$\widehat{\text{R}}_{n,\phi}^{\pi_b \to \pi_e}(z) \xrightarrow{\text{a.s.}} \text{R}_\phi^{\pi_e}(z). \tag{58}$$

*Proof.* Recall that

$$\widehat{\text{R}}_{n,\phi}^{\pi_b \to \pi_e}(z) = \frac{\sum_{i=1}^n \sum_t \mathbf{1}\{Z_t^i = z\} R_t^i \rho_{0:t}}{\sum_{i=1}^n \sum_t \mathbf{1}\{Z_t^i = z\} \rho_{0:t}}.$$

Let $X_n := \frac{1}{n} \sum_{i=1}^n \sum_t \mathbf{1}\{Z_t^i = z\} R_t^i \rho_{0:t}$ and $Y_n := \frac{1}{n} \sum_{i=1}^n \sum_t \mathbf{1}\{Z_t^i = z\} \rho_{0:t}$. Following the exact steps from Property B.3, we can show that

$$\frac{1}{n} \sum_{i=1}^n \mathbf{1}\{Z_t^i = z\} R_t^i \xrightarrow{\text{a.s.}} \mathbb{E}\left[\mathbf{1}\{Z_t = z\} R_t \rho_{0:t}; \pi_b\right]$$

$$= \mathbb{E}\left[\mathbf{1}\{Z_t = z\} R_t; \pi_e\right]$$

$$= \sum_{r \in \mathbb{R}, \check{z} \in \mathcal{Z}} \Pr(R_t = r, Z_t = \check{z}; \pi_e)\mathbf{1}\{Z_t = z\}r$$

$$= \sum_{r \in \mathbb{R}} \Pr(R_t = r, Z_t = z; \pi_e)r$$

$$= \left(\sum_{r \in \mathbb{R}} r \Pr(R_t = r \mid Z_t = z; \pi_e)\right) \Pr(Z_t = z; \pi)$$

$$= \mathbb{E}\left[R_t \mid Z_t = z; \pi\right] \Pr(Z_t = z; \pi_e).$$

This convergence guarantee holds for all $t$. By the countable additivity property of probability measures, in conjunction with the dominated convergence theorem, this implies that

$$X_n \xrightarrow{\text{a.s.}} \sum_t \mathbb{E}\left[R_t \mid Z_t = z; \pi_e\right] \Pr(Z_t = z; \pi_e). \tag{59}$$

Similarly, these same exact steps can be followed to show that

$$Y_n \xrightarrow{\text{a.s.}} \sum_t \Pr(Z_t = z; \pi_e). \tag{60}$$

Noting that $\widehat{R}_{n,\phi}^{\pi_b \to \pi_e} = \frac{X_n}{Y_n}$, we can apply the continuous mapping theorem, as done in the proof of Property B.3, to show that

$$\widehat{R}_{n,\phi}^{\pi_b \to \pi_e}(z) \xrightarrow{\text{a.s.}} R_\phi^{\pi_e}(z), \tag{61}$$

when $\sum_t \Pr(Z_t = z; \pi_b) \neq 0$. $\qquad \square$

**Property B.8.** *For all abstract states $z \in \mathcal{Z}$, $\widehat{\eta}_{n,\phi}^{\pi_b \to \pi_e}(z) \xrightarrow{\text{a.s.}} \eta_\phi^{\pi_e}(z)$.*

*Proof.* Recall that

$$\widehat{\eta}_{n,\phi}^{\pi_b \to \pi_e}(z) = \frac{1}{n} \sum_{i=1}^n \mathbf{1}\{Z_0^i = z\}\rho_0.$$

Let $X_n := \frac{1}{n} \sum_{i=1}^n \mathbf{1}\{Z_0^i = z\}\rho_0$. Following the exact steps from Property B.4, we can show that

$$\frac{1}{n}\mathbf{1}\{Z_t^i = z\} \xrightarrow{\text{a.s.}} \mathbb{E}\left[\mathbf{1}\{Z_0 = z\}\rho_0\right] = \Pr(Z_0 = z).$$

This proof holds for all $z \in \mathcal{Z}$, not just the abstract states that have non-zero probability of occuring. This implies that

$$\widehat{\eta}_{n,\phi}^{\pi_b \to \pi_e}(z) \xrightarrow{\text{a.s.}} \eta_\phi^{\pi_e}(z). \tag{62}$$

$\qquad \square$

**Lemma 3.3.** *Under Assumption 2.1, the weighted maximum likelihood estimate $\widehat{\mathfrak{R}}_{n,\phi}^{\pi_b \to \pi_e}$ converges almost surely to the ground-truth ARP $\mathfrak{R}_\phi^{\pi_e}$, i.e., $\widehat{\mathfrak{R}}_{n,\phi}^{\pi_b \to \pi_e} \xrightarrow{\text{a.s.}} \mathfrak{R}_\phi^{\pi_e}$.*

*Proof.* The results required to establish this result are provided in Properties B.6, B.7, and B.8. Specifically, the properties establish that if

$$\sum_t \Pr(Z_t = z; \pi_b) \neq 0,$$

then

$$\widehat{P}_{n,\phi}^{\pi_b \to \pi_e}(z, z') \xrightarrow{\text{a.s.}} P_\phi^{\pi_e}(z, z'); \quad \widehat{R}_{n,\phi}^{\pi_b \to \pi_e}(z) \xrightarrow{\text{a.s.}} R_\phi^{\pi_e}(z); \quad \widehat{\eta}_{n,\phi}^{\pi_b \to \pi_e}(z) \xrightarrow{\text{a.s.}} \eta_\phi(z),$$

giving the final result $\widehat{\mathfrak{R}}_{n,\phi}^{\pi_b \to \pi_e} \xrightarrow{\text{a.s.}} \mathfrak{R}_\phi^{\pi_e}$, while the values of the components for $z \in \tilde{Z}$ are hardcoded as detailed in Property B.6.

$\qquad \square$

## B.4 Off-Policy Evaluation with ARPs

**Theorem 4.1.** *The expected return of the ARP $\widehat{\mathfrak{R}}_{n,\phi}^{\pi_b \to \pi_e}$ (built from off-policy data) converges almost surely to the expected return of $\pi_e$, i.e., $J(\pi_e; \widehat{\mathfrak{R}}_{n,\phi}^{\pi_b \to \pi_e}) \xrightarrow{\text{a.s.}} J(\pi_e; M)$.*

*Proof.* The expected return $J(\pi_e; \widehat{\mathfrak{R}}_{n,\phi}^{\pi_b \to \pi_e})$ is a continuous function of $\widehat{\mathfrak{R}}_{n,\phi}^{\pi_b \to \pi_e} :=$ $(\widehat{P}_{n,\phi}^{\pi_b \to \pi_e}, \widehat{R}_{n,\phi}^{\pi_b \to \pi_e}, \widehat{\eta}_{n,\phi}^{\pi_b \to \pi_e})$, given by $J(\pi_e; \widehat{\mathfrak{R}}_{n,\phi}^{\pi_b \to \pi_e}) = \left(\mathrm{I} - \widehat{P}_{n,\phi}^{\pi_b \to \pi_e}\right)^{-1} \widehat{R}_{n,\phi}^{\pi_b \to \pi_e} \widehat{\eta}_{n,\phi}^{\pi_b \to \pi_e}$. Similar to the proof of Lemma 3.2, by the continuous mapping theorem and Lemma 3.3, we obtain the final result that enables off-policy evaluation with ARPs:

$$J(\pi_e; \widehat{\mathfrak{R}}_{n,\phi}^{\pi_b \to \pi_e}) \xrightarrow{\text{a.s.}} J(\pi_e; \mathfrak{R}_\phi^{\pi_e}) = J(\pi_e; M).$$

$\qquad \square$

## B.5 Variance Reduction: Leveraging *Markovness* of the State Abstraction

Variance in estimation of the model of the ARP from off-policy can be reduced by clipping the importance weights. We now show that when the state abstraction function is $c$-th order Markov, clipping the importance weights to the $c$ most recent terms continues to provide a consistent off-policy expected return estimate.

**Theorem 4.3.** *Given a $c$-th order Markov $\phi$, the expected return of the abstract reward process $\widehat{\mathfrak{R}}_{\phi,c}^{\pi_b \to \pi_e}$ converges almost surely to the expected return of $\pi_e$, i.e., $J(\pi_e; \widehat{\mathfrak{R}}_{\phi,c}^{\pi_b \to \pi_e}) \xrightarrow{a.s.} J(\pi_e; M)$.*

*Proof.* We only need to show that for a state abstraction function $\phi \in \Phi$ that is $c$-th order Markov, i.e.,

$$\Pr(\phi(S_{t+1})|\phi(S_t), \phi(S_{t-1}), \ldots, \phi(S_{(t-c+1)^+}); \pi) = \Pr(\phi(S_{t+1})|\phi(S_t), \phi(S_{t-1}), \ldots, \phi(S_1), \phi(S_0); \pi),$$

for $\pi \in \{\pi_b, \pi_e\}$ the following holds:

$$\widehat{\mathfrak{R}}_{\phi,c}^{\pi_b \to \pi_e} \xrightarrow{a.s.} \mathfrak{R}_\phi^{\pi_e}.$$

The remaining steps to show that the return estimates are consistent follow from Theorem 4.1. For brevity, we denote by $Z_t := \phi(S_t)$ for the rest of the proof. Consider the expression for the transition function without any weight clipping,

$$\widehat{P}_{n,\phi}^{\pi_b \to \pi_e}(z, z') := \frac{\sum_{i,t} \mathbf{1}_t^i\{z, z'\} \rho_{0:t}}{\sum_{i,t} \mathbf{1}_t^i\{z\} \rho_{0:t}}.$$

As shown in Property B.6, if $\sum_t \Pr(Z_t = z; \pi_b) \neq 0$, as $n \to \infty$ the numerator of $\widehat{P}_{n,\phi}^{\pi_b \to \pi_e}(z, z')$ converges almost surely to $\sum_t \mathbb{E}\left[\mathbf{1}\{Z_t = z, Z_{t+1} = z'\} \rho_{0:t}; \pi_b\right]$. When the abstract states are $c$-th order Markov,

$$\rho_{0:(t-c)^+} \perp \rho_{(t-c+1)^+:t}, \text{ conditioned on } (Z_i)_{i=t-c+1}^t. \tag{63}$$

Using the tower rule [60], $\mathbb{E}\left[\mathbf{1}\{Z_t = z, Z_{t+1} = z'\} \rho_{0:t}; \pi_b\right]$ can be re-written as:

$$\mathbb{E}\left[\mathbf{1}\{Z_t = z, Z_{t+1} = z'\} \rho_{0:t}; \pi_b\right]$$
$$=\mathbb{E}\left[\mathbf{1}\{Z_t = z, Z_{t+1} = z'\} \rho_{0:(t-c)^+} \rho_{(t-c+1)^+:t}; \pi_b\right]$$
$$=\mathbb{E}\left[\mathbb{E}\left[\mathbf{1}\{Z_t = z, Z_{t+1} = z'\} \rho_{0:(t-c)^+} \rho_{(t-c+1)^+:t} \mid (Z_i)_{i=(t-c+1)^+}^t; \pi_b\right]; \pi_b\right]$$
$$\overset{(a)}{=}\mathbb{E}\left[\mathbb{E}\left[\mathbf{1}\{Z_t = z, Z_{t+1} = z'\} \rho_{(t-c+1)^+:t} \mid (Z_i)_{i=(t-c+1)^+}^t; \pi_b\right] \mathbb{E}\left[\rho_{0:(t-c)^+} \mid (Z_i)_{i=(t-c+1)^+}^t; \pi_b\right]; \pi_b\right]$$
$$=\mathbb{E}\left[\mathbb{E}\left[\mathbf{1}\{Z_t = z, Z_{t+1} = z'\} \rho_{(t-c+1)^+:t} \mid (Z_i)_{i=(t-c+1)^+}^t; \pi_b\right]; \pi_b\right]$$
$$\qquad \mathbb{E}\left[\mathbb{E}\left[\rho_{0:(t-c)^+} \mid (Z_i)_{i=(t-c+1)^+}^t; \pi_b\right]; \pi_b\right]$$
$$=\mathbb{E}\left[\mathbf{1}\{Z_t = z, Z_{t+1} = z'\} \rho_{(t-c+1)^+:t}; \pi_b\right] \underbrace{\mathbb{E}\left[\rho_{0:(t-c)^+}; \pi_b\right]}_{=1}$$
$$\overset{(b)}{=} \Pr(Z_t = z, Z_{t+1} = z'; \pi_e),$$

where $(a)$ follows from Equation 63 and $(b)$ follows from Lemma 3.3. Thus when the abstract states are $c$-th order Markov, the numerator estimated with $c$-clipped importance weights almost surely converges to the same value as if the importance weights were not clipped. The value to which the denominator converges similarly undergoing a change of measure by the use of clipped importance weights, i.e.,

$$\mathbb{E}\left[\mathbf{1}\{Z_t = z\} \rho_{0:t}; \pi_b\right] = \mathbb{E}\left[\mathbf{1}\{Z_t = z\} \rho_{(t-c+1)^+:t}; \pi_b\right] = \Pr(Z_t = z; \pi_e), \tag{64}$$

when $\phi$ is $c$-th order Markov. Note that the expression for $\widehat{P}_{\phi,c}^{\pi_b \to \pi_e}$ is:

$$\widehat{P}_{\phi,c}^{\pi_b \to \pi_e}(z, z') := \frac{\sum_{i,t} \mathbf{1}_t^i\{z, z'\} \rho_{(t-c+1)^+:t}}{\sum_{i,t} \mathbf{1}_t^i\{z\} \rho_{(t-c+1)^+:t}}.$$

Let $X_n := \frac{1}{n} \sum_{i,t} \mathbf{1}_t^i \{z, z'\} \rho_{(t-c+1)^+:t}$ and $Y_n := \frac{1}{n} \sum_{i,t} \mathbf{1}_t^i \{z\} \rho_{(t-c+1)^+:t}$. We can write $\widehat{\mathrm{P}}_{\phi,c}^{\pi_b \to \pi_e}(z, z') = \frac{X_n}{Y_n}$. Following the exact steps from Lemma 3.3, that entail a careful application of the continuous mapping theorem, we can show that

$$\widehat{\mathrm{P}}_{\phi,c}^{\pi_b \to \pi_e}(z, z') \xrightarrow{\text{a.s.}} \mathrm{P}_\phi^{\pi_e}(z, z'),$$

if $\sum_t \Pr(Z_t = z; \pi_b) \neq 0$. Similar derivations can be followed for the reward function and the initial state distribution, leading to the required result:

$$\widehat{\mathfrak{R}}_{\phi,c}^{\pi_b \to \pi_e} \xrightarrow{\text{a.s.}} \mathfrak{R}_\phi^{\pi_e}$$

when $\sum_t \Pr(Z_t = z; \pi_b) \neq 0$, that enables almost sure convergence of the expected return estimate:

$$J(\pi_e; \widehat{\mathfrak{R}}_{\phi,c}^{\pi_b \to \pi_e}) \xrightarrow{\text{a.s.}} J(\pi_e; \mathfrak{R}_\phi^{\pi_e}) = J(\pi_e; M).$$

$\square$

# C   Empirical Details

In this section we provide additional empirical details for the experiments presented in the main paper. An overall step-by-step algorithm for STAR is as follows:

---

**Algorithm 1** Overview of STAR($\phi, c$)

---

**Input:** $\pi_e, \pi_b, \mathcal{D}_n^{(\pi_b)}$

1. Apply state abstraction to $\mathcal{D}_n^{(\pi_b)}$ and compute importance weights:

$$\forall\, i, t : \text{Store} \left( Z_t^{(i)} = \phi\left(S_t^{(i)}\right), A_t^{(i)}, R_t^{(i)}, \rho_{(t-c+1)^+:t}^{(i)} = \prod_{j=(t-c+1)^+}^{t} \frac{\pi_e(S_j^{(i)}, A_j^{(i)})}{\pi_b(S_j^{(i)}, A_j^{(i)})} \right)$$

2. Estimate the components of the ARP $\widehat{\mathfrak{R}}_{\phi,c}^{\pi_b \to \pi_e}$:

$$\widehat{\mathfrak{R}}_{\phi,c}^{\pi_b \to \pi_e} = \left( \widehat{P}_{\phi,c}^{\pi_b \to \pi_e}, \widehat{R}_{\phi,c}^{\pi_b \to \pi_e}, \widehat{\eta}_{\phi,c}^{\pi_b \to \pi_e} \right) \text{ where,}$$

$$\widehat{P}_{\phi,c}^{\pi_b \to \pi_e} = \frac{\sum_{i,t} \mathbf{1}\{\phi(S_t^{(i)})=z, \phi(S_{t+1}^{(i)})=z'\} \rho_{(t-c+1)^+:t}}{\sum_{i,t} \mathbf{1}\{\phi(S_t^{(i)})=z\} \rho_{(t-c+1)^+:t}},$$

$$\widehat{R}_{\phi,c}^{\pi_b \to \pi_e} = \frac{\sum_{i,t} \mathbf{1}\{\phi(S_t^{(i)})=z\} \rho_{(t-c+1)^+:t} R_t^{(i)}}{\sum_{i,t} \mathbf{1}\{\phi(S_t^{(i)})=z\} \rho_{(t-c+1)^+:t}},$$

$$\widehat{\eta}_{\phi,c}^{\pi_b \to \pi_e}(z) = \frac{\sum_{i=1}^{n} \mathbf{1}\{\phi(S_0^{(i)})=z\}}{n}$$

3. Compute the expected return $J(\pi_e; \widehat{\mathfrak{R}}_{\phi,c}^{\pi_b \to \pi_e})$ from the ARP.

$$J(\pi_e; \widehat{\mathfrak{R}}_{\phi,c}^{\pi_b \to \pi_e}) := (I - \widehat{P}_{\phi,c}^{\pi_b \to \pi_e})^{-1} \widehat{R}_{\phi,c}^{\pi_b \to \pi_e} \widehat{\eta}_{\phi,c}^{\pi_b \to \pi_e}$$

**Output:** $J(\pi_e; \widehat{\mathfrak{R}}_{\phi,c}^{\pi_b \to \pi_e})$

---

## C.1   Domains

We perform empirical evaluations on a range of domains that consist of continuous domains and domains with large state spaces with long horizons. The domains are as follows:

**CartPole**   The CartPole domain is a classic control problem from OpenAI Gym [6]. The task is to balance a pole on a cart by moving the cart left or right. The state space is continuous, and the action space is discrete. We use the standard CartPole environment from OpenAI Gym. In the experiments, $\pi_b$ is uniformly random, i.e., the left and right actions are each taken with probability 0.5. The evaluation policy $\pi_e$ is a policy that takes the action right with probability 0.9 when the pole is leaning left, and right with probability 0.1 when the pole is leaning right. This results in a policy that is not optimal, but is somewhat successful at balancing the pole.

**ICU-Sepsis**   The ICU-Sepsis domain simulates the treatment of sepsis in the ICU. Built from the MIMIC-III database [24] and drawing from the analysis of Komorowski et al. [26], it consists of 747 states that denote the status of a patient and 25 possible actions that denote possible medical interventions. At the end of each episode, if the patient survives, a reward of +1 is given, while death corresponds to a reward of 0, with all intermediate rewards also being 0. This results in the expected return of a policy corresponding to the probability that a randomly selected patient will survive. In the experiments, $\pi_e$ is set to an *expert policy*, provided with the domain and included in the submitted codebase, while $\pi_b$ is a policy that is a more stochastic version of the expert policy constructed by temperature scaling [17] the action probabilities of expert policy with a temperature parameter $\tau = 2$.

**Asterix**   The Asterix domain is a miniaturized version of the Atari game Asterix where the task is to collect items while avoiding enemies. We use the implementation of the game from the MinAtar testbed [61], where the dimension of each state is $10 \times 10 \times 4$, and the action set consists of six actions. The data collecting policy $\pi_b$ is uniformly random, while the evaluation policy $\pi_e$ picks actions with non-uniform skewed probabilities.

## C.2 Additional Results for Estimator Selection

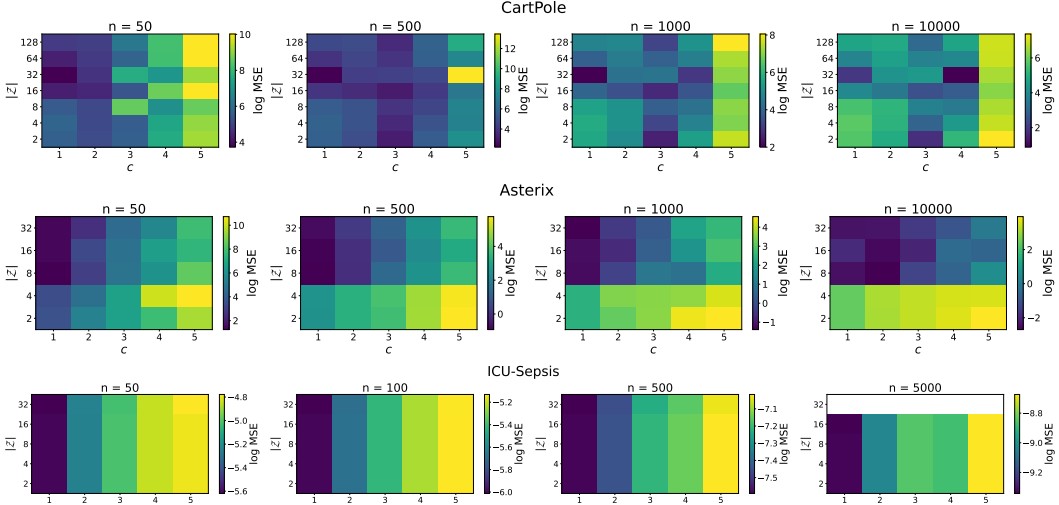

Figure 4: Heatmap of the log mean squared error (MSE) of the OPE estimators for the CartPole, Asterix, and ICU-Sepsis domains. The vertical axis represents the number of abstract states $|\mathcal{Z}|$, and the horizontal axis represents the value of the hyperparameter $c$. The color intensity indicates the log MSE, with lower values denoting better performance. Note that the variation with $|\mathcal{Z}|$ is strongly influenced by the class of abstraction functions used, which in this work is CluSTAR.

For each domain, we evaluate the OPE performance of the estimated ARP induced by varying configurations of $(\phi, c)$. For the class of abstraction function, we observe that the simple method CluSTAR performs well across all domains, and hence we use it for all experiments. CluSTARtakes an input a single hyperparameter, the number of centroids initialized, denoted by $|\mathcal{Z}|$. We evaluate the following configurations of $\mathcal{Z}$ and $c$ for each domain:

1. CartPole: 35 estimators - $|\mathcal{Z}| \in \{2, 4, 8, 16, 32, 64, 128\}$, $c \in \{1, 2, 3, 4, 5\}$.
2. ICU-Sepsis: 25 estimators - $|\mathcal{Z}| \in \{2, 4, 8, 16, 32\}$, $c \in \{1, 2, 3, 4, 5\}$. ICU-Sepsis with $|\mathcal{Z}| = 32$ and $n = 5000$ is excluded for computational reasons.
3. Asterix: 25 estimators - $|\mathcal{Z}| \in \{2, 4, 8, 16, 32\}$, $c \in \{1, 2, 3, 4, 5\}$.

In Figure 4 we report the log MSE for each estimator.

**Effect of $|\mathcal{Z}|$:** For the class of abstraction functions induced by CluSTAR, the effect of $|\mathcal{Z}|$ varies across domains. In some domains, such as ICU-Sepsis, the estimators obtained by varying this hyperparameter show similar performance. However, in domains like Asterix and CartPole, performance is more sensitive to the choice of $|\mathcal{Z}|$, with larger values performing better.

**Effect of $c$:** It is expected that large values of $c$ will lead to higher variance of the estimates. In low data regimes, small values of $c$ result in relatively lower MSE as the variance is lowered at the cost of increased bias. As the amount of data increases, the best value of $c$ also increases. This effect is most pronounced in the Asterix domain, and to a lesser extent in the CartPole domain.

The key takeaway from Figure 4 is that the optimal $|Z|$ and $c$ are a function of the amount of data ($n$), the characteristics of the domain and the class of abstraction functions used.

## C.3 Compute

All experiments were run for 200 seeds each, on 3 domains in total. Each run took between 3 hours to 3 days (depending on the domain) and this duration includes offline data collection. The experiments

were run using 32 threads on Xeon E5-2680 CPUs on a computing cluster, bringing the total compute time to roughly 45000 compute hours.

