# OpenReview forum: "Abstract Reward Processes: Leveraging State Abstraction for Consistent Off-Policy Evaluation"
_NeurIPS.cc/2024/Conference — NeurIPS 2024 poster_

### Official Review · Reviewer_5Mtw · 2024-07-11

**Soundness:** 3
**Presentation:** 3
**Contribution:** 3
**Rating:** 7
**Confidence:** 3

**Summary:**

The authors introduce a new framework that groups the states with a similar reward, hence reduces variance of any applied OPE estimator.

**Strengths:**

The framework is technically sound, well discussed, the analysis is on point and addresses important questions, for example that regardless of the choice of state abstraction function \phi, the overall framework remains asymptotically consistent. The discussion on how the framework translates to existing OPE methods when varying its parameters (\phi, c) help show where the improvements of this framework lie.

The experiments are complete and show a strong empirical performance of this method.

**Weaknesses:**

Minor: [L548, L560, L586, L600, L609] "See Appendix B" is obsolete, we are already in Appendix B.

**Questions:**

In [L212] you claim this is the first model-based OPE method that converges to a true policy value without any model class assumptions. How does your method compare to action clustering method of Peng et al. (2023)? This method factors action space instead of the state space, and regardless of the model class, it seems to converge to a true policy value.

In [L304], you suggest that performance can be improved even further by applying estimator selection methods. Moreover, in [L250], you mention there is no approach to discovering good abstraction functions \phi. The recent work of Cief et al. (2024) might solve both these issues.

References:

Peng, Jie, Hao Zou, Jiashuo Liu, Shaoming Li, Yibao Jiang, Jian Pei, and Peng Cui. “Offline Policy Evaluation in Large Action Spaces via Outcome-Oriented Action Grouping.” In Proceedings of the ACM Web Conference 2023, 1220–30. WWW ’23. New York, NY, USA: Association for Computing Machinery, 2023. https://doi.org/10.1145/3543507.3583448.

Cief, Matej, Michal Kompan, and Branislav Kveton. “Cross-Validated Off-Policy Evaluation.” arXiv, May 24, 2024. http://arxiv.org/abs/2405.15332.

**Limitations:**

The authors extensively discuss the limitations throughout the work, for example, in Conclusions.

---

> ### Author Rebuttal · Authors · 2024-08-06
>
> We thank you for taking time to review our paper and appreciate you recognizing the strength of the work’s theoretical and empirical analysis. We appreciate you pointing out additional relevant references, and address related questions below.
>
>
> > In [L212] you claim this is the first model-based OPE method that converges to a true policy value without any model class assumptions. How does your method compare to action clustering method of Peng et al. (2023)? This method factors action space instead of the state space, and regardless of the model class, it seems to converge to a true policy value.
>
> GroupIPS performs IPS on the clustered actions and is thus a model-free method, whereas our work involves approximating a model of the system, specifically, an abstract Markov reward process. The use of discrete state abstractions, in conjunction with tabular models, frees our method from requiring knowledge about the model class that can represent the underlying MDP. While the referenced paper does not seem to provide a guarantee for convergence to the true policy value (as per Proposition 3.3, the method is biased), our approach offers the theoretical guarantee of asymptotic convergence to the true policy value.
>
> > In [L304], you suggest that performance can be improved even further by applying estimator selection methods. Moreover, in [L250], you mention there is no approach to discovering good abstraction functions \phi. The recent work of Cief et al. (2024) might solve both these issues.
>
> Thank you for highlighting this reference on estimator selection for OPE. Holding out some off-policy data to estimate a proxy of the true policy value, using a validation OPE estimator, is a promising approach to abstraction selection. We will incorporate this reference and discuss its relationship to our work in the updated version of the paper.
>
> >  [L548, L560, L586, L600, L609] "See Appendix B" is obsolete, we are already in Appendix B.
>
> We shall remove that text in the updated version of the paper.
>
> We hope we have addressed your concerns, kindly let us know if you have any additional questions.

---

> ### Comment · Reviewer_5Mtw · 2024-08-07
>
> Thank you for taking the time to answer my questions. After reviewing all discussions, I am keeping my score and will advocate for accepting the paper (if the discussion will be needed, it seems the vote is unanimous).

---

### Official Review · Reviewer_V4gu · 2024-07-11

**Soundness:** 3
**Presentation:** 3
**Contribution:** 3
**Rating:** 7
**Confidence:** 2

**Summary:**

The paper introduces a new framework, STAR, for off-policy evaluation (OPE). OPE uses a chain with rewards (MRP) to conduct the evaluation. The challenge of this methodology is that the MRP estimation can introduce bias, given the shift between the behavior policy and the policy for evaluation. STAR is designed to address this challenge. STAR projects continuous state space to a tabular space and introduces the abstract reward process (ARP), which refers to the MRP over an abstract state space. The tabular abstract state space reduces the difficulty of reward prediction, thus eliminating the model class mismatch in prediction. It is proved in the paper that projecting states to a discrete space preserves the return.

**Strengths:**

* The paper makes good contributions by proposing a new framework for off-policy evaluation. The proposed method is analyzed in detail with both mathematical proofs and empirical evaluations.

* The introduction clearly lists the challenges and contributions, as well as provides a subsection for an overview of the new method. The way the introduction is organized tunes to be helpful for better understanding.

* For the introduced step, abstracting continuous states with a tabular space, the paper explains the advantages clearly. The paper also provides mathematical proof to explain why this step is not considered as a source of error.

**Weaknesses:**

* Theorem 3.1 proves why state abstraction preserves the return with a known reward function on the abstraction state space, but it remains unclear how the state abstraction increases the difficulty of learning the transition model for predicting the reward and the next state, and possibly also introduces stochasticity. Assuming the true MDP has $P(r_0, s_1| s_0, a_0) = 1$ and $P(r_1, s_3|s_1, a_0)=1$, if the state abstraction projects $s_0$ and $s_1$ to the same representation $\phi_0$, which is possible because the latent space is tabular. Then, the model will have to simulate a stochastic transition with probability $P(r_0, s_1 | \phi_0, a_0)=0.5$ and $P(r_1, s_3 | \phi_0, a_0)=0.5$. Therefore, it seems how accurate the evaluation is heavily depends on how good the state abstraction is. If so, it would be good to point out in the paper.

**Questions:**

Please see the concern above

**Limitations:**

Yes

---

> ### Author Rebuttal · Authors · 2024-08-06
>
> We thank you for taking the time to review our paper, and appreciate your recognition of our work's contributions and analyses. Your observation regarding differing ease of model estimation across various abstractions is insightful, as detailed next. We have updated the paper to include this point.
>
> > Theorem 3.1 proves why [...] in the paper.
>
> As correctly pointed out, in some cases abstraction by aggregation of states can increase the difficulty of estimation of the transition function (ref: `Note A`). However, in general, state aggregation tends to simplify estimation by increasing the effective sample size [1].
>
> `Note A:` For example, aggregation of two states with deterministic transitions -- which can be estimated perfectly from a single observation of those transitions -- creates stochastic transitions between abstract states. For these states, the abstraction function being the identity mapping is preferable to one that is a many-to-one mapping.
>
> As per Theorems 3.1 and 4.1, MSE for OPE using STAR converges to zero as the amount of data increases, regardless of the abstraction function used. However, the rates of convergence can vary with the choice of abstraction. This highlights the need to consider additional factors, such as the ease of estimation and dataset size, while selecting an abstraction in STAR.
>
> We appreciate you pointing out this subtlety and hope we addressed your concern. We have updated the paper to include this point - kindly let us know if you have any additional questions.
>
> ---
>
> [1] Jiang, Nan. "Notes on state abstractions." (2018) URL: https://nanjiang.cs.illinois.edu/files/cs598/note4.pdf

---

> > ### Comment · Reviewer_V4gu · 2024-08-12
> > **Reply**
> >
> > I would like to thank the authors for their reply. The reply addressed my concern. After reading the reply and other reviews, I increased the score.

---

### Official Review · Reviewer_oq5d · 2024-07-12

**Soundness:** 4
**Presentation:** 4
**Contribution:** 3
**Rating:** 8
**Confidence:** 3

**Summary:**

This paper studies the problem of Off-Policy Evaluation (OPE), which consists of estimating the value of a policy $\pi_e$ from an input dataset generated from another behaviour policy $\pi_b$. One naive way of constructing the estimated return of $\pi_e$ would be to compute its associated empirical Markov Reward Process (MRP) in the MDP, then apply importance sampling to it. Instead, the technique proposed in the paper is to apply importance sampling to an abstract MRP obtained via some discrete abstraction function. This technique has been validated with theoretical results, and with an experimental comparison against other baselines.

**Strengths:**

(Idea and contribution) The core idea, that is marginalizing states into abstract MRP, is simple. However, it requires special care in selecting appropriate abstract representations that allow lossless evaluation of arbitrary policies. As confirmed by Theorem 3.1, this is indeed the case for the ARP defined in the paper (though it should be noted that this strongly relies on $\gamma = 1$). From Theorem 3.1 all the other results follow, which verify that the proposed technique is sound for *any mapping function*. The fact that "actions propagate through the abstract state transitions" (line 227) is also a very interesting addition, that contributes to the application of importance sampling. Summarizing, the theory developed in the paper is an elegant display of the abstraction process for OPE, and a sound use of HRL principles.

(Evaluation) The technique has been compared with a good number of relevant baselines from the literature. The results show that the proposed estimators have competitive or better performance than their competitors. The authors also reported the median estimator. Even though it is not always the best estimator, this addition allows to have a clearer picture, and it was evaluated positively.

(Presentation) The quality of presentation is high. Each new technical definition is well motivated, and the paper is reasonably self contained.

(Reproducibility) The authors provided the full source code at the time of submission. The hyperparameters are also provided.

**Weaknesses:**

1. The theoretical results show that the method is sound and any estimator constructed as explained in the paper is an unbiased estimate of the true value.
However, the paper does not provide any theoretical result regarding the reduction in variance, which is the second motivation of this work. This is only confirmed experimentally in some domains.

1. No finite samples analysis has been conducted. Thus, when considering datasets of finite size, no theoretical result regarding the approximation error is available.

1. The method assumes to explicitly know the behaviour policy that generated the data. This is a strong limitation that has been also listed by the authors in the limitation section. This did not contribute heavily to the final score.

Minor:
- There are some typos in the use of z and z' in equation (2) and in the proof of Theorem 3.1.
- If accepted, I suggest to use the additional page for explaining the missing details of the experimental evaluation, such as:
  - How is the MSE for OPE exactly defined
  - How is the variance-bias measure is constructed for Figure 3.
  - List Algorithm 1 in the main body

**Questions:**

4. How did you obtain the probabilities of the behaviour policy for the ICU sepsis dataset?
5. Which of the baselines assume complete knowledge about the behaviour policy?

The authors may also address any of the points raised above.

**Limitations:**

The authors listed all the most important limitations of this work. The contribution has no direct societal impact and it does not require further discussion regarding negative impact.

---

> ### Author Rebuttal · Authors · 2024-08-06
>
> We thank you for your careful review of our paper and insightful comments. We appreciate your positive assessment of our contributions and the clarity of our presentation. We address your questions and comments below.
>
> > How did you obtain the probabilities of the behaviour policy for the ICU sepsis dataset?
>
> The behavior policy is constructed by increasing the temperature parameter to the logits of an expert policy, making it more stochastic. The expert policy weights are provided provided with the ICU-Sepsis benchmark. Behavior probabilities are obtained by querying this modified behavior policy.
>
> > Which of the baselines assume complete knowledge about the behaviour policy?
>
> We thank the reviewer for pointing out the shared assumption of a known behavior policy among the baselines. All baselines, other than fitted Q-evaluation (FQE), assume knowledge of the behavior policy. Variants to some methods have been proposed where the behavior policy is empirically estimated [1], introducing some bias in estimation, and we expect these to be a straightforward practical extension to STAR.
>
> > Theoretical proof for variance reduction
>
> The framework of STAR encompasses a range of estimators, controlled by $(\phi, c)$, each configuration of which instantiates a specific estimator. The efficacy of variance reduction of these estimators is inherently linked to the problem of *abstraction discovery*. Different combinations of abstractions and weight clipping factors yield varying degrees of variance reduction compared to standard OPE, with some configurations potentially not offering any reduction. For instance, weighted per-decision importance sampling (WPDIS) and approximate-model MRP (AM-MRP), representing the end points within STAR (ref: Section 4.2), cannot, in general, be ordered by their variance. Similarly, estimators that lie in between may exhibit significantly lower or higher variance and MSE. We assert the *existence* of such low variance estimators and provide empirical support for this claim.
>
> > Finite sample analysis
>
> A finite sample analysis would necessitate an extension of the simulation lemma [2] for ARPs defined from finite-horizon MDPs. Subsequently, the policy value estimation error can be bounded in terms of the estimation errors in the transition and reward functions of the estimated ARP. Such an extension to the simulation lemma would be a valuable theoretical contribution independent of its application to our work, and represents a promising direction for future exploration.
>
> > Minor
>
> Thank you for the suggestions, we shall incorporate these points in the additional space in the updated version.
>
> We hope we have addressed your concerns, kindly let us know if you have any additional questions.
>
> ---
>
> [1] Hanna, Josiah, Scott Niekum, and Peter Stone. "Importance sampling policy evaluation with an estimated behavior policy." International Conference on Machine Learning. PMLR, 2019.
>
> [2] Kearns, Michael, and Satinder Singh. "Near-optimal reinforcement learning in polynomial time." Machine learning 49 (2002): 209-232.

---

### Public Comment · ~Shreyas_Chaudhari1 · 2026-04-08
**Comment by Authors | Errata**

We would like to highlight that in its current form Theorem 4.3 as stated in in our paper [Chaudhari et al., 2024] is false. Specifically, one step in its proof does not hold, invalidating the proof. We note this error along with a suggested corrected in the follow [errata document](https://shreyasc-13.github.io/docs/ARP_errata.pdf).  The rest of the paper and its theoretical and empirical results remain unchanged.

---

### Decision · Program_Chairs · 2024-09-25

**Decision:**

Accept (poster)

**Comment:**

This paper presents an intriguing new framework, STAR, for consistent model-based off-policy evaluation (OPE) using abstract reward processes (ARPs). Theorem 3.1 is quite an interesting building block of the paper, allowing estimation of value under state abstraction. Reviewers agreed this paper is strong, and raised some concerns. A few points about the paper (not all points were raised by reviewers):

- The authors use tabular models for representing ARPs. While they argue that even small models can be effective, there is no analysis of how the complexity scales with the size of the original state space or the desired accuracy. For continuous state spaces, the computational cost of constructing the state abstraction is also not discussed.
- While the paper provides asymptotic consistency results, it lacks finite-sample analysis. This makes it difficult to assess the practical performance of STAR for finite datasets. Finite-sample error bounds would be crucial for understanding how the estimation error depends on factors like the number of trajectories, the horizon length, the state abstraction function, and the weight clipping factor.
- The authors acknowledge that there is no principled method for selecting the weight clipping factor c. A theoretical or empirical analysis of how to choose c for different ARPs and dataset sizes would be beneficial.
- Although the authors claim that STAR often inherits favorable properties of both IS-based and model-based methods, they do not provide analysis of the bias-variance trade-off. How does the choice of abstraction and weight clipping factor affect the bias and variance of the ARP estimates? How does this trade-off compare to existing OPE methods?
- The authors should improve the overall related work section to include recent advances in OPE. There is vast amount of research on the combination of model and IS based approaches for OPE, contrary to what the authors claim in their paper.

Overall the authors have a novel and interesting result in the paper, though they do not go all the way in term of the analysis, e.g., finite sample analysis and / or bias variance bounds. This makes it difficult to theoretically compare to related work. The paper's empirical evaluation is good and greatly improves the quality of the paper. The paper is still in a borderline state, and I lean toward accepting it. I urge the authors to try to improve the paper as best they can for the final submission given the above comments and the other reviewer comments.